# Homogentisic acid-derived pigment as a biocompatible label for optoacoustic imaging of macrophages

Ina Weidenfeld[1], Christian Zakian[1,2], Peter Duewell[3], Andriy Chmyrov [1,4], Uwe Klemm[1], Juan Aguirre[1,2], Vasilis Ntziachristos[1,2,4] & Andre C. Stiel [1]*

Macrophages are one of the most functionally-diverse cell types with roles in innate immunity, homeostasis and disease making them attractive targets for diagnostics and therapy. Photo- or optoacoustics could provide non-invasive, deep tissue imaging with high resolution and allow to visualize the spatiotemporal distribution of macrophages in vivo. However, present macrophage labels focus on synthetic nanomaterials, frequently limiting their ability to combine both host cell viability and functionality with strong signal generation. Here, we present a homogentisic acid-derived pigment (HDP) for biocompatible intracellular labeling of macrophages with strong optoacoustic contrast efficient enough to resolve single cells against a strong blood background. We study pigment formation during macrophage differentiation and activation, and utilize this labeling method to track migration of pro-inflammatory macrophages in vivo with whole-body imaging. We expand the sparse palette of macrophage labels for in vivo optoacoustic imaging and facilitate research on macrophage functionality and behavior.

---

[1] Institute of Biological and Medical Imaging (IBMI), Helmholtz Zentrum München, Neuherberg, Germany. [2] Chair of Biological Imaging, Technische Universität München, Munich, Germany. [3] Institute of Innate Immunity, University of Bonn, Bonn, Germany. [4] Center for Translational Cancer Research (TranslaTUM), Technische Universität München, Munich, Germany. *email: andre.stiel@helmholtz-muenchen.de

As an essential component of the innate immune system, macrophages play a pivotal role during host defense against pathogens, in tissue maintenance by clearance of apoptotic cells, as well as during various inflammatory processes in the body caused by infection, allergic reaction or disease. Increasing evidence even points to macrophage involvement in autoimmune diseases, such as rheumatoid arthritis, inflammatory bowel disease, multiple sclerosis, and asthma (reviewed in ref. [1]).

It is important to not only investigate the molecular activation and signaling steps that regulate macrophage functionality, but also to understand how these steps play out in a temporal and spatial manner. Macrophages are found throughout the body, and their blood-associated (monocytic) precursors become highly mobile upon stimulation by specific chemical cues[2]. These cells are recruited to sites of infection or injury, where they phagocytose exogenous invaders or endogenous debris to launch appropriate cellular responses. They can cross endothelial barriers as easily as spread throughout tissues, adapting their level of differentiation and activation accordingly[3,4]. Their motility and morpho-functional plasticity make them a major challenge for biomedical imaging. Current methodologies to study macrophages in vivo include, e.g., Bioluminescence Imaging (BLI) or Intravital Microscopy (IVM) often in conjunction with nanoparticles (NPs)[5]. Limited by poor spatial resolution and short tracking duration (BLI), or low penetration depths, imprecise cell labeling and decreased animal survival (IVM) (reviewed in ref. [6]), there is a strong need for non-invasive, whole-body imaging methods that allow longitudinal studies under physiological conditions.

Recently emerged optoacoustic (OA, also termed photoacoustic) imaging offers high-resolution visualization of optical contrasts in tissues at depths well below the 1 mm limit typical of optical microscopy. In particular, this technique resolves optical absorption and is therefore sensitive to different tissue chromophores such as hemoglobin or melanin. The method already shows strong applications[7,8] in visualizing tissue oxygenation[9], inflammation[10,11], and metabolism[12] based on resolving intrinsic tissue hemoglobin. However, non-invasive tracking of activated macrophage subpopulations is still greatly limited by the sparse availability of biocompatible cellular labels that attain appropriate optoacoustic contrast, i.e., high optical absorption and non-radiative energy dissipation. So far, several dyes, transgene-encoded pigments (melanin, violacein, X-Gal) or chromophore-bearing proteins (reviewed in ref. [13]) as well as plasmonic nanomaterials or NPs have been considered (reviewed in refs. [14,15]). While stable transgenic labels would be advantageous for longitudinal studies, macrophages are difficult to transfect and sensitive to genetic alterations[16], which may result in silencing of the transgene or toxicity due to uncontrollable overexpression. Many protein labels do not provide signal-to-noise ratios sufficient to visualize small cell numbers, much less single cells, or they are sensitive to photobleaching resulting in rapid signal loss. Exogenous contrast agents such as (in)organic dyes or—nanomaterials have strong OA absorbance but must be administered with caution due to cytotoxicity and possible perturbation of cellular performance (reviewed in refs. [17–20]). Further, the systemic administration of diagnostic or therapeutic nanomaterials used to target cells in vivo are often associated with poor delivery (<5%), as nanomaterials likely accumulate in filtrating organs such as the liver, kidneys, spleen, or lungs[21]. Misguided immune reactions and inflammation are often the consequence. Non-specific hydrophobic dye-based markers[22] show rapid loss of signal, often within hours, in addition to carry-over of the label to neighboring cells or tissues (M. Kimm, personal communication).

Among the above-mentioned contrast agents, melanin has emerged as an interesting pigment for optoacoustics. With a very strong absorbance that extends far into the near infrared (NIR) it ensures high contrast in tissues, even in the presence of hemoglobin in the vasculature[23], an essential prerequisite for macrophage tracking. Naturally occurring across the majority of species, from humans to microbes, the melanins comprise a large family of bio-macromolecules generated in higher organisms through enzymatic oxidation and polymerization of tyrosine or produced in lower organisms through auto-oxidation of phenolic compounds. The predominant form present in humans is brown-black eumelanin, and while it mainly serves as a cellular photoprotectant it is also responsible for the unique coloration of skin, hair, and eyes. Eumelanin, like most melanins, is a highly insoluble polymer found restricted to specialized intracellular compartments of endogenously pigmented cells[24,25]. Thus far, eumelanin produced via transgenic tyrosinase catalysis has been utilized for optoacoustic studies in different mammalian cell lines[23,26,27]. However, while this pigment shows strong optoacoustic signal, cell viability is greatly affected by random intracellular enrichment of these large insoluble polymers whose synthesis additionally produces toxic intermediates. Therefore, despite proof of principle demonstrations, eumelanin-producing cell systems did not find broad biological imaging applications. Recently, more efforts are being made to synthesize melanin-containing contrast agents with better solubility, thus, higher cell tolerance and promising application (reviewed in ref. [28]).

Based on this, we aimed to identify an entirely biocompatible labeling method that overcomes the insolubility and cytotoxicity problems of eumelanin while preserving cell functionality so that adept optoacoustic imaging of cells in tissues is enabled. We noted that in the genetic disorder alkaptonuria[29], deficiency of the enzyme homogentisate 1,2-dioxygenase[30] leads to the formation of black pigmented plaques preferentially located on collagen fibers in connective tissues of cartilage and bone[31,32]. This enzyme participates in the phenylalanine/tyrosine catabolism, and its loss-of-function leads to accumulation of the small molecule homogentisic acid (HGA). Decades of continuous HGA buildup in patients' serum, urine, and interstitial fluids are required for alkaptonuria to manifest and result in the characteristic ochronotic pigment deposits primarily found in joints. Cartilage degeneration[33] and SAA amyloid–pigment coaggregations have further been reported[34].

In several fungi and bacteria, HGA is also present and is natively converted into the only known soluble melanin species—pyomelanin, which in this case has a cell-protective function[35,36]. While ochronotic pigment in alkaptonuria is insoluble and its detailed chemical structure and size are still to be determined, HGA-based pyomelanins, as well as pure HGA-pigment in solution, form small polymers of 10–20 kDa via HGA auto-oxidation followed by oligomerization into soluble brown-black pigment[37,38].

We therefore hypothesized that treatment of macrophages with HGA will trigger the intracellular production of a highly soluble HGA-derived pigment (referred to as HDP), similar to pyomelanin, however, different from ochronotic pigment. We further conjectured that HDP production could offer strong contrast for optoacoustics while ensuring biocompatibility.

The present study is the first to demonstrate the OA properties of HDP in vitro and in vivo and to establish its easy application for intracellular labeling of primary and immortalized macrophages. We show the development of an OA labeling method even gentle enough to apply during primary macrophage differentiation and activation. Unlike other exogenous labeling processes, the present approach does not require that cells phagocytose large pre-synthesized materials; instead, the HDP label is generated in situ in living cells, thereby greatly reducing stress and cytotoxicity while ensuring high OA signal. HDP

proves to have no detectable intrinsic activity in macrophages. Concurrently, it preserves cellular functions such as chemically induced polarization and innate motility. As a proof of concept, we use mammalian HDP to track the migration of pro-inflammatory macrophages to a site of simulated inflammation in vivo. In addition, HDP allows single-cell macrophage visualization in blood deploying raster-scan optoacoustic mesoscopy (RSOM). With this we hope to facilitate macrophage monitoring as an OA method for various areas of basic and pre-clinical research where the involvement of these versatile immune cells is not yet fully understood.

## Results

**Characterization of HGA-derived pigmentation in vitro.** A considerable advantage of working with a soluble small molecule such as HGA (Fig. 1a) in mammalian cell culture is the easy preparation of a sterile stock solution and the straightforward

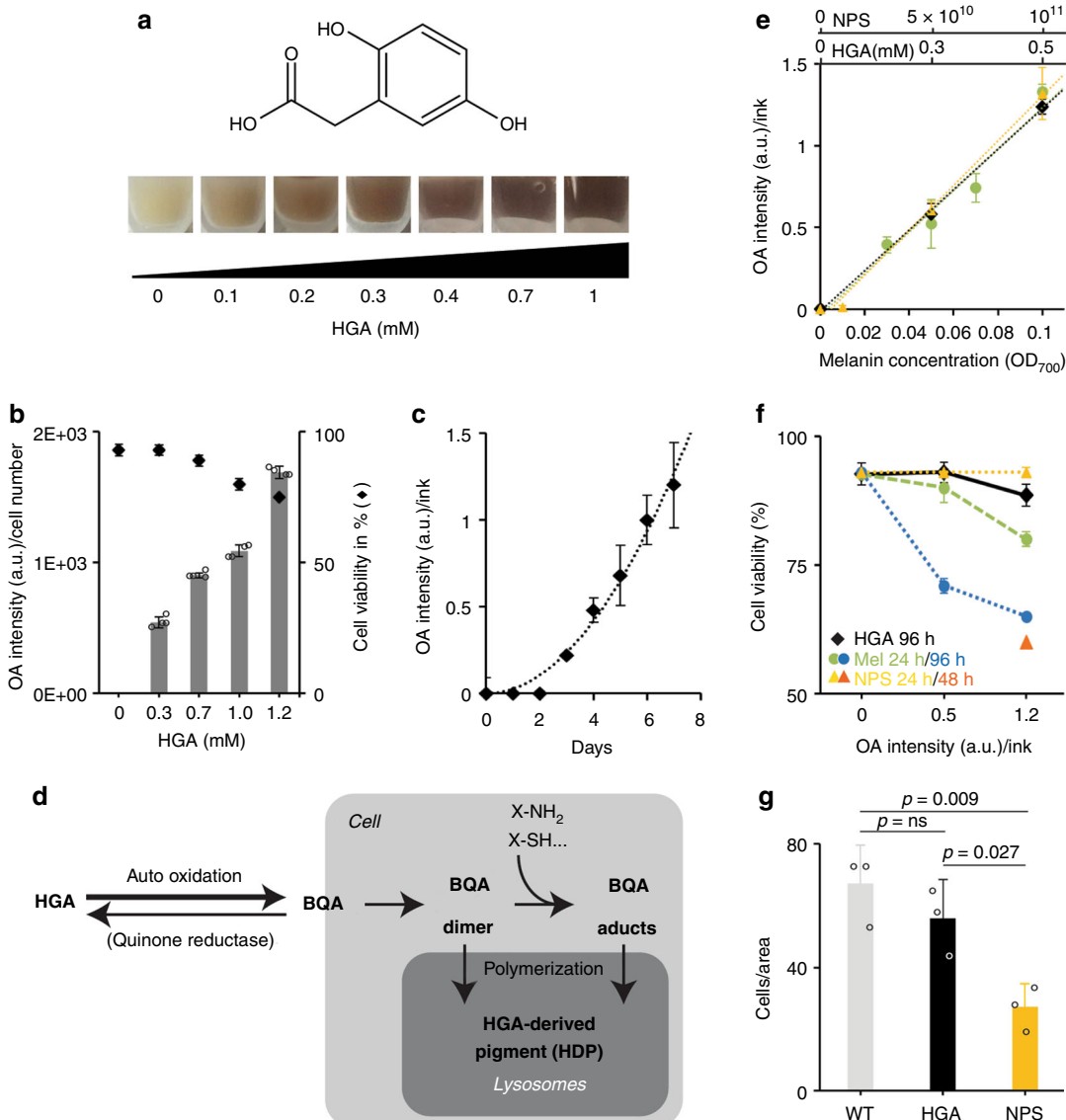

**Fig. 1** OA- and physiological characterization of HGA-derived pigmentation in vitro. **a** Chemical structure of homogentisic acid (HGA, top). Pelleted Ana-1 cells showing increasing pigmentation after treatment with 0–1 mM HGA for 96 h (bottom). **b** Normalized optoacoustic signal intensities of Ana-1 cells plotted against their corresponding HGA concentrations (0, 0.3, 0.7, 1, 1.2 mM) after 96 h of treatment. The secondary axis shows the percentage of viable cells for each concentration (black diamond). $N = 4$. **c** Polymerization kinetics of the pigment followed by a single dose of 0.3 mM HGA to Ana-1 cells shown as a function of OA signal. $N = 2$, $z = 7$. **d** Hypothetical model of OA-sensitive HDP formation in macrophages. Our data implies that the formation of HDP occurs at an accelerated rate in the presence of intact cells indicated by the reaction taking place inside the box (Cell). We do not exclude reactions additionally occurring at other locations. **e–g** Comparison of HDP vs. synthetic eumelanin and nanoparticles (NPs; SiAuRods) in macrophages. **e** Linear increase of OA signal intensities measured in macrophages as a function of concentration of HGA (0, 0.3, 0.5 mM), of the soluble fraction taken from synthetic melanin ($OD_{700nm}$ of 0, 0.03, 0.05, 0.07, 0.1) and of supplemented NPs (0, $5 \times 10^{10}$, $10^{11}$ particles). HDP and synthetic eumelanin were recorded at 700 nm and NPs at 780 nm. $N = 2$–6, $z = 6$. **f** Cell viability plotted against the OA signal of macrophages treated with HGA for 96 h, synthetic melanin for 24 h and 96 h, or NPs for 24 h and 48 h. $N = 2$–9. **g** In vitro cell motility of primary M1-polarized macrophages comparing untreated (WT) with HPD vs. NPs-labeled cells. Cells were pretreated with HGA for 96 h and NPS for 16 h prior to start and allowed to migrate for 24 h. Motility of NPs-fed cells was significantly decreased compared with HPD and WT cells, the latter two showing no significant (ns) difference. $N = 3$. Error bars, mean ± SD. $N$ = biological replicates, $z$ = number of positions for data acquisitions within a replicate. Source data are provided as a Source Data file

application of a desired concentration. In contrast, insoluble synthetic melanin needs several weeks to partially solubilize in hydrophilic solution and can then only be quantified by its spectral absorbance. Further, many NPs for NIR imaging are not optimized for efficient cell uptake and their quantification can be challenging due to particle aggregation, which consequently leads to a reduction in OA signal.

In order to determine the intracellular pigmentation achievable upon HGA treatment, different concentrations of the HGA stock solution were directly added to the growth media of the representative macrophage cell line Ana-1. We could adjust the polymerization level of HDP by adding HGA to the medium at final concentrations of 0.1–1 mM (Fig. 1a). To estimate the potential of the macrophage label in OA imaging, intracellular pigment concentrations were directly quantified based on their OA signal obtained by the multispectral optoacoustic tomography system (MSOT). As depicted in Fig. 1b, normalized optoacoustic signal at 700 nm showed a linear increase as a function of HDP concentration. Similar results were also obtained by adding HGA to cultures of J774A macrophages or primary bone marrow-derived macrophages.

Cell viability was assessed after traditional Trypan Blue staining followed by scoring the ratios of viable to apoptotic cells. We refrain from using the colorimetric MTT assay due to a possibility of signal interference from the HDP. We observed no loss of cell viability at concentrations of up to 0.3 mM of HGA, with a nonsignificant decrease of viability measured at 0.5 mM HGA. This is presumably due to the greater solubility and smaller size of HDP compared with other melanin pigments[36]. High HGA concentrations of 0.7–1 mM still allowed 80% viable cells (Fig. 1b), while final concentrations above 1 mM HGA showed even stronger pigmentation but reduced cell viability below 80% and are hence not recommended.

We examined the polymerization kinetics of HGA to HDP to determine the time point of detectable OA signal and visible pigmentation in macrophages. OA signal and brown pigmentation of cells is first detected after 3 days of incubation with a single initial treatment of 0.3 mM HGA (Fig. 1c). Up to day 7, we observe a linear increase in OA signal likely caused by the gradual polymerization of HDP in vivo resulting in brown-black coloration of cells with minimal to no reduction of cell viability. In the following we employ standard HDP-labeling conditions with 0.3 mM HGA and 96 h of incubation to ensure that the cells were still in their exponential growth phase during experimentation.

**Cellular pathways for HDP formation in macrophages**. We wanted to explore whether this oligomerization of HDP occurred outside the macrophages or within the cells. Two basic routes for intracellular pigment enrichment can be envisioned: (i) cellular uptake of HGA monomers with intracellular polymerization possibly in highly acidic[39] vesicular compartments such as lysosomes[40], or (ii) extracellular formation of polymer intermediates followed by cellular ingestion via fluid phase uptake. With HGA being a small polar molecule, and a derivative of the amino acid tyrosine, it can be hypothesized to diffuse directly across the lipid bilayer of membranes as well as potentially be transported, like tyrosine, across the membrane by $Na^+$-independent L or T transport systems[41,42]. Our results (Supplemental Note 1 and Supplemental Fig. 1a–c) suggest that HDP formation is primarily intracellular and occurs after HGA uptake. To probe the potential role of the intracellular environment in promoting HGA polymerization, we followed polymerization kinetics in growth media at stable neutral pH and over 96 h in the presence and absence of Ana-1 cells as well as in presence of cellular debris by absorbance

spectroscopy (Supplemental Note 1 and Supplemental Fig. 2). We notice an increase in absorbance starting at 600 nm and persisting to 900 nm only in the presence of intact cells. We speculate that this absorbance peak corresponds to the HDP responsible for the OA signals we detect in cells at 630 nm (RSOM) and >680 nm (MSOT). We hypothesize, therefore, that HDP forms from HGA via 1,4-benzoquinone-2-acetic acid (BQA). Consistent with this idea, HGA and BQA are known to form a redox system with a highly positive redox potential of BQA[43] which consequently leads to the formation of homo-/heterodimers and small oxidized polymers thereof[36]. These results strongly implicate intracellular events in HDP formation. We conclude that macrophages prefer to take up HGA and small BQA intermediaries after which HDP —generating strong OA signal—gradually forms within the cell. By conducting bright field, fluorescence and high-resolution OA microscopy, HDP was found localized to intracellular vesicles identified to be endolysosomal organelles where high acidity might facilitate polymerization and HDP formation (Fig. 2 and Supplemental Note 2). Since all melanins are highly heterogenous polymers often incorporating inorganic as well as organic materials[38], it is conceivable that cellular materials are integrated during polymerization (Fig. 1d). Further details of this highly interesting pigmentation process remain to be elucidated and might provide insight into the molecular processes involved in alkaptonuria.

**OA properties and viability of macrophages labeled with HDP**. To further assess the HGA-based labeling system for macrophages, we performed a basic comparative study with synthetic (eu-)melanin as well as commercially available silica-coated 10 nm gold nanorods, the latter two requiring active phagocytosis. HDP shows similarities to pyomelanin[38], which in turn has an absorption spectrum similar to that of eumelanin[44]. To compare their respective OA properties and cytotoxicity in vitro, Ana-1 cells as well as primary bone marrow-derived macrophages were treated with HGA or synthetic eumelanin. As a consequence of the low solubility of synthetic melanin, a PBS-based solution was prepared 3 weeks in advance. The soluble fraction thereof was recovered after centrifugation and filter-sterilized before use. We treated the same numbers of cells for 24 or 96 h with different concentrations of the soluble melanin fraction ($OD_{700nm}$ = 0, 0.03, 0.05, 0.07, and 0.1, quantified by absorbance spectroscopy as described above) or with different amounts of HGA (0, 0.3, 0.5, 0.8 mM), respectively. In parallel, cells were incubated with increasing numbers of NPs for 24 h. Figure 1e shows the linear increase of OA signal as a function of concentration for both melanins (700 nm) and for NPs (780 nm). For comparison, cellular pigmentation generated with either 0.5 mM HGA, synthetic melanin with $OD_{700nm}$ 0.1 or $10^{11}$ supplemented NPs (calculated to 30,000/cell) gave rise to nearly identical OA signal of ~2.0. Higher concentrations of HGA led to even stronger OA signals, and these signals could not be reached using synthetic melanin because of its limited solubility in aqueous solution. Absorption measurements of supplemented media at start vs. end of incubation showed that NPs uptake remained below 10,000 particles/cell in our hands. Cell viability was significantly lower with synthetic melanin at $OD_{700nm}$ 0.1 than with 0.5 mM HGA, while NPs uptake did not alter viability within 24 h and with increasing concentrations. High cell toxicity, however, was observed after 48 h of NPs treatment (Fig. 1f). Further, cytotoxicity was observed when Ana-1 cells were exposed to (i) synthetic melanin $OD_{700nm}$ 0.1 for 96 h (viability at 70% and below), (ii) the insoluble fraction of synthetic melanin for 24 h (viability decreased to 50%), or (iii) synthetic melanin in DMSO (Supplemental Fig. 3). When we dissolved

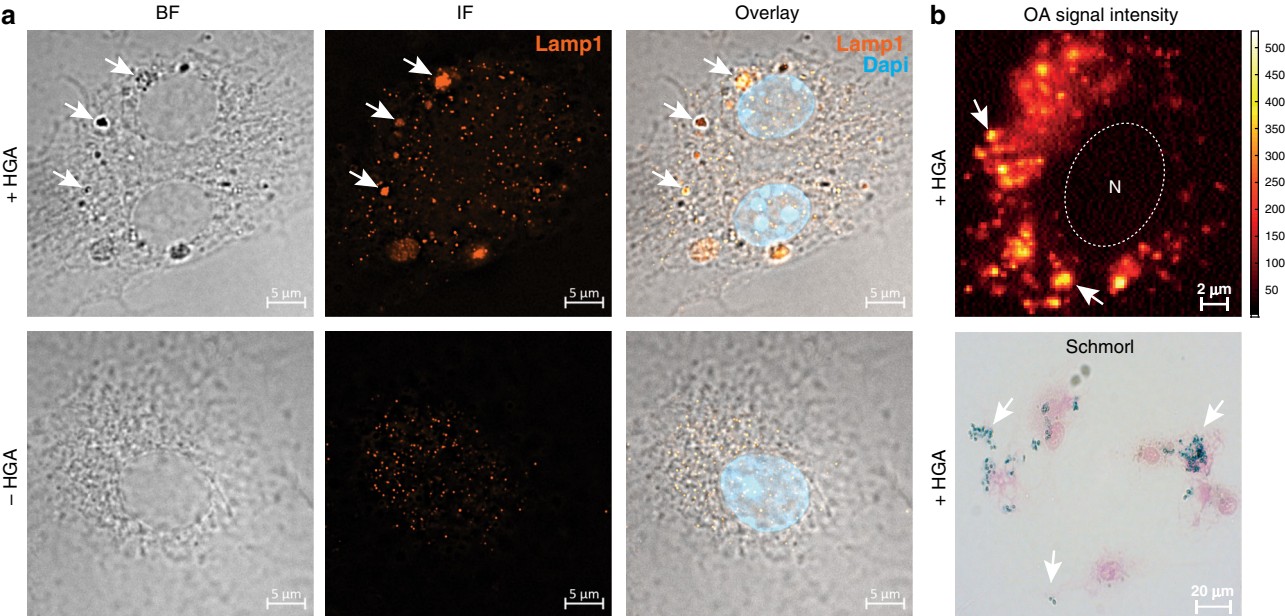

**Fig. 2** Microscopic confirmation of intracellular HDP formation in primary macrophages after HGA treatment. **a** Intracellular co-localization of HDP-laden vesicles, as seen in bright field, with late endosomes/lysosomes verified by immunofluorescence staining with anti-Lamp1 antibody (Alexa 488, orange). Arrows indicate single pigmented vesicles and aggregates thereof (top arrow pointing at aggregate). Cell nuclei are stained with Dapi. Scale bar is 5 μm. **b** Live cell optoacoustic signal acquisition of HDP allowing intensity quantification of single vesicles surrounding the cell nucleus indicated by N (top, pixel size is 0.2 μm, scale bar of 2 μm). HDP vesicles are visualized by blue coloration after Schmorl's staining of cells (bottom, scale bar is 20 μm). +HGA cells were treated with 0.5 mM HGA for 96 h

synthetic melanin in DMSO, we obtained strong OA signals with solutions of $OD_{700nm}$ 0.2–0.4, but cell viabilities decreased to 60% after 24 h and 48% after 96 h, even though the final DMSO concentration never exceeded 1%. In addition to elevated levels of cell mortality, synthetic melanin exposure induced morphological alterations in macrophages. As seen in Supplemental Fig. 4d, ca. 50% of the cells detached from the surface and formed large floating aggregates, a phenomenon not observed in untreated cultures or cultures treated with HGA.

With cell viability routinely monitored during the course of HGA exposure, it is equally important to assess this factor after depletion of the supplement with regard to longitudinal in vivo application of pre-labeled cells. Synthetic melanin exposure takes a toll on cells to such an extent that viability drops below 75% even after 72 h of recuperation in label-free growth media. As for HGA treatment, we observe a significantly smaller effect on cell viability with values at 89% (Supplemental Fig. 4b and Supplemental Note 3). In summary, HDP shows equally strong OA properties as the soluble fraction of phagocytosed eumelanin as well as the NPs tested herein. HDP, however, does not impact longevity or morphology of labeled macrophages which are essential requirements for functional in vivo studies.

**Verification of macrophage functionality after HDP-labeling.**
In preparation for in vivo experiments, it is important to confirm that HGA-mediated macrophage labeling does not compromise plasticity and function of cells, in order to preserve the macrophage's differentiation capacity. To address this capacity to differentiate under HGA (Supplemental Fig. 5 and Supplemental Note 4), total bone marrow cells from FoxN1 nude mice were incubated with 0.5 mM HGA from day 2 until day 5 of differentiation (day 1 being the day of extraction) resulting in black coloration of cells with 98% cell viability (Supplemental Fig. 5b). In addition, cells were treated with HGA for days 5–8 of differentiation, immunostained with APC-conjugated anti-F4/80 and

FITC-conjugated anti-CD11b[45] and subjected to fluorescence-activated cell sorting (FACS) to confirm that more than 90% of the population had efficiently transformed into primary macrophages during HGA treatment (Supplemental Fig. 6). In terms of infection, macrophages are recruited and activated to initiate an adequate immune response, e.g., by eradicating pathogens. HGA-pigmented cells were therefore treated with bacterial lipopolysaccharide (LPS) for 24 h on days 9–10[46]. A CD38-based flow cytometry assay was capable of identifying positive subpopulations with 93% of the HGA-labeled cells, compared with 83% of non-labeled macrophages, being highly activated macrophages (Supplemental Fig. 6)[47].

To further ensure that HGA/HDP does not perturb primary macrophage functionality, we measured the secretion of 24 different cytokines and chemokines in the presence and absence of HGA as well as with or without LPS (representatives shown in Supplemental Fig. 7b). HGA alone did not show intrinsic activation of macrophages, with no signs of pro- or anti-inflammatory responses, such as IL-1b, IL-6, TNFa, or IL-10 and IL-13, respectively. In addition, other cytokines and chemokines also remained unaltered. Most importantly, HGA did not change the macrophages response toward LPS challenge, confirming our inert HDP-based labeling method. An LDH release assay further confirmed that HDP is non-toxic in the present cell system, even after 5 days of HGA incubation (Supplemental Fig. 7a).

By performing an in vitro motility assay we confirmed that HDP-laden primary macrophages, which were polarized to an M1-like phenotype by using LPS, preserved their ability to migrate to a comparable degree as their unlabeled counterparts (Fig. 1g). In contrast, NPs-laden cells, although unaffected in their viability, demonstrated significantly reduced motility. The above results suggest that the usage of gradual in vivo/in situ labeling methods via HGA, for primary cells such as macrophages, has advantages over the feeding of cells with pre-synthesized particles such as, e.g., synthetic eumelanin or the tested nanorods, which can be challenging for subsequent in vivo applications.

In this respect we tested a further primary cell line, and several tissue-derived lines (Supplemental Note 4), for their degree of tolerance towards HGA-labeling. Mouse embryonic stem (mES) cells received 0.3 mM of HGA for 96 h. As depicted in Supplemental Fig. 8 these cells show intracellular pigmentation, when compared with the untreated culture, with no reduction of cell viability caused by HGA supplementation. Increasing concentrations, however, reduce viability and are not recommended.

A salient features of macrophage function is the presentation of antigen to T-cells (reviewed in ref. [48]). Taking place at close proximity, we demonstrate the lack of intercellular transfer of HDP from macrophages to another type of cell that might be recruited by the presence of IFN-γ and LPS (Supplemental Note 4 and Supplemental Fig. 9).

With our present observations we can exclude that (a) the OA signals detected in vivo originate from cells or tissues other than the injected labeled macrophages, and (b) transfer of free HGA or HDP to other tissues has occurred. To evaluate a potential inflammatory role for HDP-labeled macrophages in vivo, we assessed serum amyloid a (SAA) levels[49]. Measuring $n = 3$ mice, tail vein injected with $0.6 \times 10^6$ HDP-labeled macrophages, shows a mild (10-fold) increase in SAA serum levels at 24 h (3.24 ± 4.1 μg/ml) and 48 h (4.08 ± 2.1 μg/ml), as compared with base line (0.52 ± 0.1 μg/ml). This increase is most likely related to the tail vein injection procedure, as comparable observations have been made with FoxN1 nude mice[50] during other small interventions.

**Functional in vivo recruitment of HDP-labeled primary macrophages.** To prove the in vivo application potential, different numbers of HDP-pigmented primary macrophages were mixed with matrigel® and subcutaneously injected in FoxN1 nude mice followed by signal acquisition with multispectral optoacoustic tomography (MSOT) (Fig. 3a–c). Increasing cell numbers correspond to an increasing OA signal suggesting in vivo quantification of HGA-pigmented macrophages. Histological tissue sections stained with Schmorl's solution to highlight the HDP label confirm the presence of the macrophages distributed

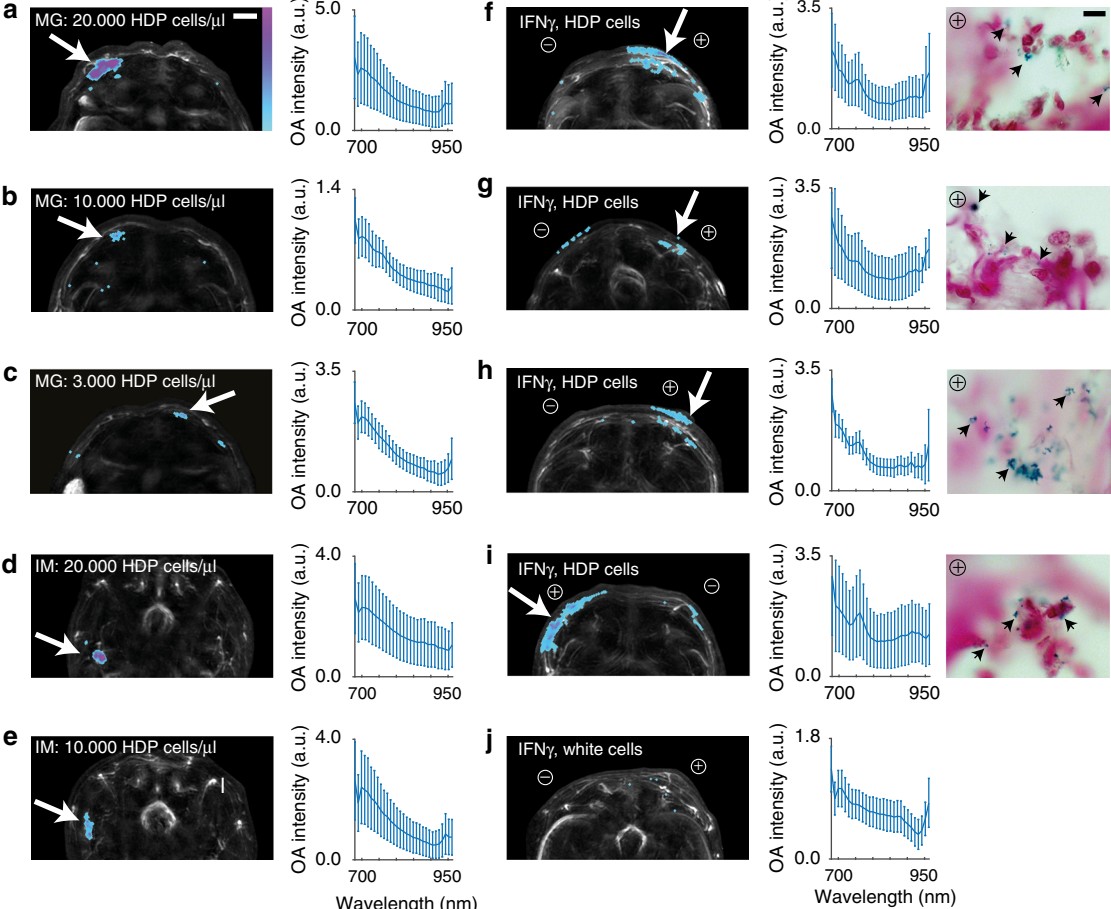

**Fig. 3** Functional in vivo recruitment of HDP-labeled primary macrophages. **a–c** MSOT signals of defined cell concentrations after subcutaneous (s.c.) implantation with matrigel®. Separation of HDP induced signal from blood within a highly vascularized organ is demonstrated by deep intramuscular injection (**d**, **e**) in a mouse hindlimb. **f–i** MSOT images of the dorsal half of FoxN1 nude mice harboring s.c. matrigel® implants enriched with Interferon-gamma (+IFNγ) or without (−IFNγ). Shown are projections of the tomography slices covering the matrigel® implants after 24 h. **j** Image of a mouse injected with unlabeled cells after 24 h. Areas that are positively unmixed for HDP-pigment shown as sum projection, colorbars denote positive unmixed pixels (see Methods) and anatomical details shown at 720 nm as average projection. Measurements prior to injection are shown in Supplemental Fig. 12. The mean spectra of the areas positively unmixed for HDP are presented adjacent, alongside a histological view of the ⊕ IFNγ implant highlighting the accumulation of the injected HDP-labeled cells (see Supplemental Figures 10 and 14). The green-blue coloration of HDP-filled intracellular vesicles is attributed to Schmorl's staining of the tissue sections. We note that the MSOT signal of the direct injections (**a–e**) is more localized and shows a cleaner melanin-like spectrum, while for the recruitments (**f–i**) we see a more disperse signal with a spectral signature presumably convoluted with blood spectra due to the migration of the labeled macrophages towards IFNγ via the circulatory system. Scale bar for MSOT images is 2 mm, for histological images, 10 μm

throughout the gel in corresponding densities (Supplemental Fig. 10a). To demonstrate strong signal recognition before a background of blood a deep injection in the highly vascularized hindlimb skeletal muscle of a mouse was performed (Fig. 3d, e). This result was further strengthened by the reproducible quantification of HDP-laden cells submerged in blood-agar phantoms compared with PBS-agar phantoms (Supplemental Fig. 11).

In order to investigate the feasibility of the present macrophage label for longitudinal monitoring/cell tracking, we setup a functional in vivo recruitment experiment. To that end primary M1-like macrophages were generated from bone marrow extract differentiated for 8 days and activated by addition of LPS for an additional 24 h. HDP was initiated during differentiation and for a total of 96 h. In total, $6 \times 10^5$ labeled—or unlabeled cells were injected in the tail vein of FoxN1 nude mice in four independent experiments. Each mouse carried two dorsal subcutaneous matrigel® implants, one of each supplemented with the cytokine Interferon-γ (IFN-γ) plus LPS to stimulate macrophage recruitment. Using MSOT we tracked labeled-macrophage migration collecting data 24 h post tail vein injection. Figure 3f–i shows intensity projections of the reconstructed images of the dorsal area covering the implants of the four animals unmixed for the spectral signature of melanin (see Methods section for details) indicating the successful recruitment of labeled macrophages to IFN-γ—positive implants after 24 h as seen by the enrichment of an HDP-melanin signal in this area. The spectra of these regions are presented adjacent. In contrast, we do not detect significant signal in the region of the IFN-γ-negative matrigel® implants of these mice, nor is there any melanin-like signal recorded prior to cell injection, or in the control animals which were injected with unlabeled cells (Fig. 3j) or PBS (Supplemental Fig. 12). Prolonged signal stability of this label is indicated by a further study where signal of HDP-laden M1-like macrophages was still detected at a +IFN-γ site 48 h post implantation of $1 \times 10^6$ labeled cells (Supplemental Fig. 13).

The localization of HDP-laden macrophages to positive recruitment sites was further confirmed by extensive histology. As evident in Supplemental Fig. 14, the darker contrast of HDP-laden macrophages is enriched in the matrigel® of the +IFNγ implants compared with the –IFNγ sites of the 4 recruitment animals (Fig. 3f–i). At higher resolution, single HDP vesicles within labeled cells are more prominently visible due to blue-green staining of melanin-pigment with Schmorl's solution.

Taking a closer look at the distribution of signal-inducing macrophages exposes a correlation to OA-signal patterning in the dorsal area (Fig. 4). Histological cross sections of the animals revealed a strong infiltration of F4/80-positive macrophages in the subdermal areas immediately above the positive recruitment implant (Fig. 4a) and, for one animal, likewise in the flanking area (Fig. 4c, bottom arrow), as compared with the negative implants. Whereas F4/80 also detects endogenous macrophages, Schmorl's staining confirms the recruitment of the injected HDP-labeled cells not only in the positive subdermis, but additionally within the positive matrigel® implants as highlighted in Fig. 4b–d. Thus, higher resolution imaging of +IFNγ implants confirms HDP-labeled macrophages surrounding hair follicles of the subdermis, at the interface of subdermis and matrigel® and embedded within the given gel (Fig. 4b, right). Neither microscopic—nor macroscopic analyses of IFNγ-positive vs. negative implant areas revealed any apparent anatomical differences such as, e.g., tissue necrosis or hemorrhaging (Supplemental Fig. 10b).

**Single-cell visualization in blood with RSOM.** We assessed if the strong OA signal of HDP in macrophages is sufficient to visualize single cells against a strong background of hemoglobin. To this end we mixed HDP-labeled primary macrophages, treated with 0.5 mM

HGA for 96 h, in sheep blood (final 25%) and low melting agar (final 1.5%) to generate an ≈ 2-mm-thick polymer droplet on which we implemented the raster-scan optoacoustic mesoscopy (RSOM) technique, frequently used for research in clinical dermatology[10]. Images were taken at 630 nm, since at that wavelength the absorption coefficient of blood is low, while that of melanin is still high. Due to the high axial resolution of RSOM down to 15 μm we could readily identify sparsely scattered single cells against the strong blood background. Figure 5a shows a volumetric scattered plot of detected events within the droplet. Here the image voxel is ~645 μm³ in volume which is below the cell volume expected average (4188 μm³ for a 20 μm-diameter cell). The detected events follow a partial Gaussian distribution with a mean diameter value of 27 μm and a peak diameter at 20 μm (Supplemental Fig. 15d), exactly matching the average cell diameter of primary macrophages in solution. We therefore prove that the HDP label renders single cells visible. Few events (12%) were detected with sizes greater than 40 μm which suggest a small proportion of aggregation of closely adjacent cells in the droplet. The blood-phantom control without cells (Supplemental Fig. 15a) shows no optoacoustic signal compared with the presence of signal for blood-agar phantom droplets with HDP-laden primary macrophages added at 2000 and 10,000 cells/μl (Supplemental Fig. 15b, c). In an in vivo experiment the above cells were subcutaneously injected in the back of the mouse through a catheter needle. Comparison of images before and after injection displays distinctive signal clusters at the exit of the catheter needle (Fig. 5b–e). The signals correspond to the labeled cells which are pushed out of the needle into the subcutaneous connective tissue layer leading to their line-up at depths of 0.7–1 mm from the epidermis surface. This further demonstrates that tracking of small HDP-labeled macrophage populations is indeed feasible with different OA methods.

## Discussion

For the majority of diseases, including various cancers and inflammatory disorders, we are well aware of the involvement of macrophages. However, many of their specific functions, e.g., cell migration or switching of activation states during onset, progression and manifestation of a given pathophysiological state are not yet fully understood. Non-invasive, OA imaging serves as a highly promising method to study inflammation by in vivo monitoring of activated macrophage migration events under physiological conditions.

For this purpose, we have established a biocompatible OA contrast agent for the easy and efficient labeling of primary bone marrow-derived macrophages. The presented cell labeling technique utilizes the small molecule HGA, which spontaneously auto-oxidizes to soluble polymers leading to intracellular pigmentation that we refer to as HDP. While this class of pigment is related to the soluble pyomelanins found in some microorganisms, its exact structure and composition as a macrophage label remains to be determined. It is conceivable that other compounds, carrying free amine or sulfhydryl groups, may be incorporated during intracellular HDP polymerization. However, this labeling strategy excludes equivalence to the etiological agent of alkaptonuria.

In contrast to labeling cells with insoluble synthetic melanin, nanoparticles or dyes, HDP is synthesized in vivo/in situ during the labeling process of macrophages. While ensuring strong OA signal, HDP's high solubility leads to the better tolerance in cells suggesting a high biocompatibility.

The present labeling process proved to be gentle enough to be applied to isolated bone marrow cells during or after macrophage differentiation while ensuring complete cell viability. Biocompatibility further includes the label to have no detectable intrinsic activity as seen by unaffected cytokine/chemokine levels, however,

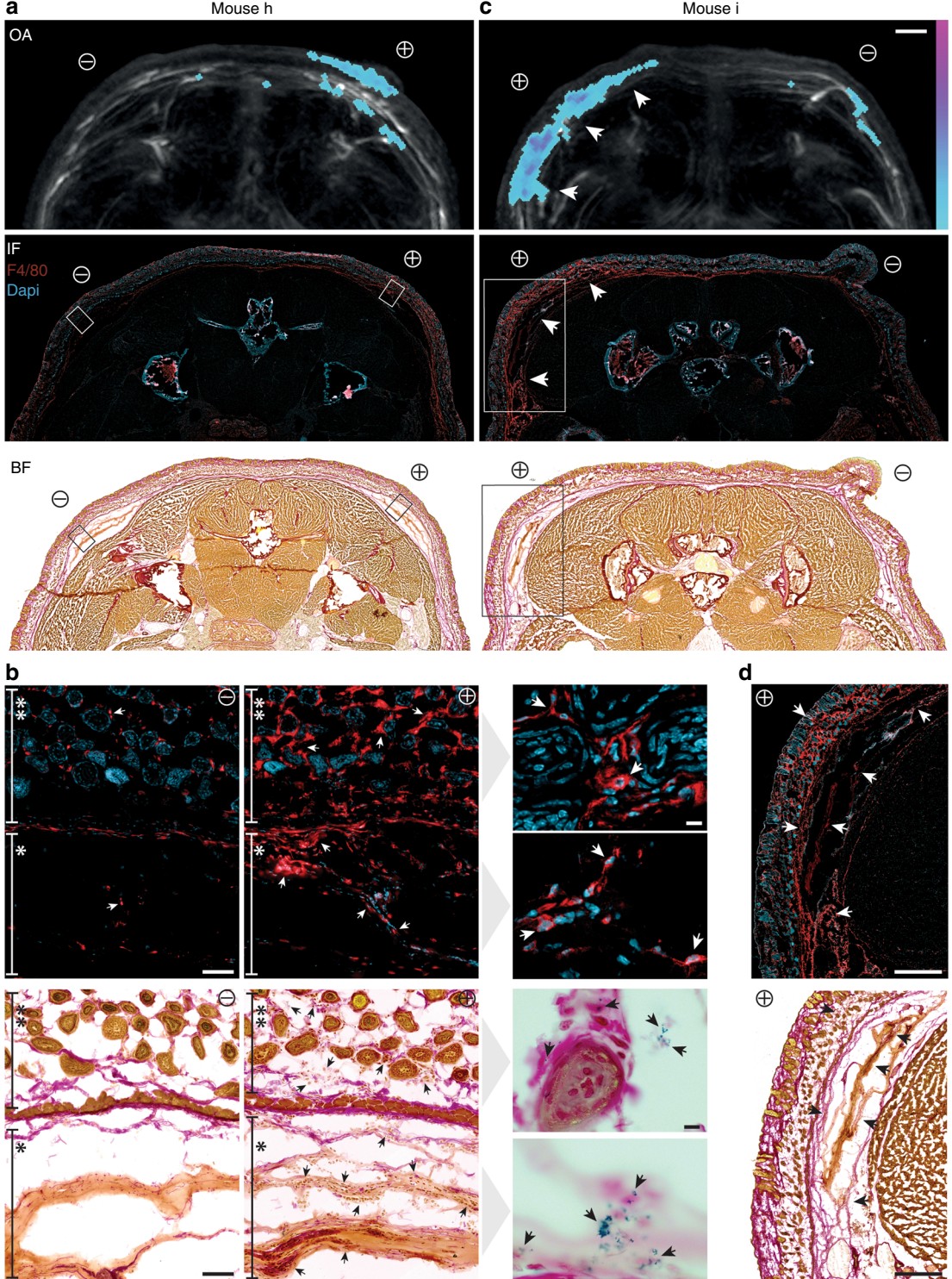

**Fig. 4** Confirmation of in vivo OA signal distribution by histology. **a, c** OA signal distribution generated by HDP-laden macrophages in MSOT (see Fig. 3, top) correlating to immunohistochemical (IHC) staining with the macrophage marker F4/80 (red, middle) and Schmorl's staining, imaged in bright field (BF, bottom), to identify labeled macrophages in histological tissue sections (10 μm) of these animals. All signals accumulate in the area of +IFNγ implants (+) versus −IFNγ implants (−). Boxes in (**a**) and (**c**) are shown at high resolution in (**b**) and (**d**), respectively. The strong infiltration of macrophages is seen in the skin and subdermal tissue areas (**b; d**) as well as within the matrigel® (*b; **c** and **d** arrows) of +IFNγ-implants. In (**c**) there is additional signal found in the area flanking the +IFNγ implant (bottom arrow) which can be accounted for (**d**). Single macrophages are indicated by white arrows in IHC images and black arrows in Schmorl's-BF. A 100x objective identifies those macrophages carrying HDP-laden vesicles (black arrows) which are found surrounding hair follicles in the skin and residing within the matrigel® of positive implants (**b**, right). Cell nuclei are stained by Dapi in IHC. Scale bars are 2 mm (**a, c**), 100 μm and 10 μm (**b**), and 600 μm (**d**).

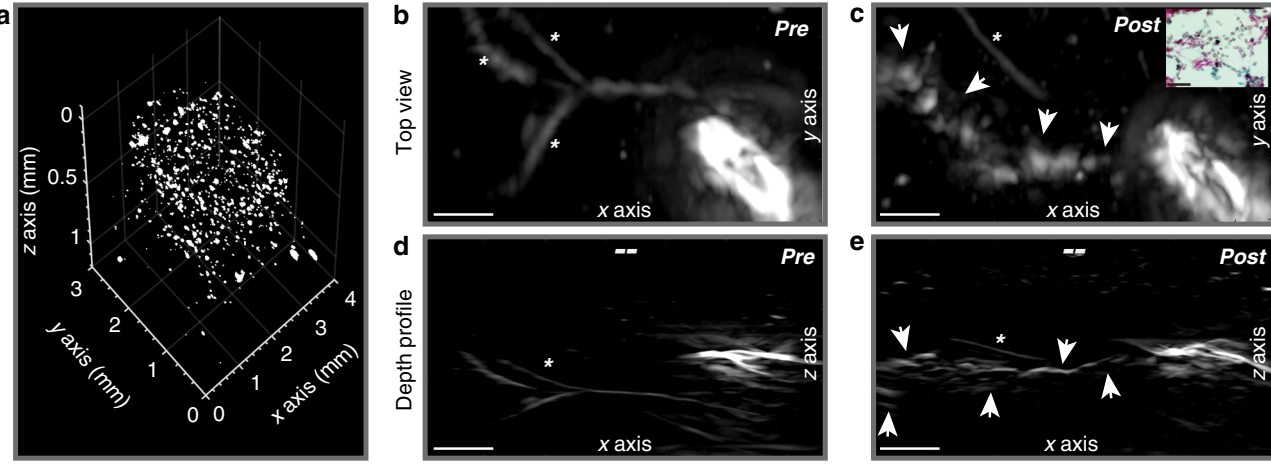

**Fig. 5** HDP facilitates single-cell visualization with raster-scan optoacoustic mesoscopy (RSOM). **a** Signals of HDP-laden primary macrophages are separated from hemoglobin in blood-agar phantoms and depicted in a volumetric scatter plot. The detected events follow a partial Gaussian size distribution with a peak diameter of 20 μm, compliant with the average diameter of primary macrophages in solution (see Supplemental Fig. 15). **b**–**e** Subcutaneous injection in the dorsal area of a FoxN1 nude mouse of the cells measured in (**a**). A catheter was used to determine the injection area and scans were recorded pre- (**b**, **d**) and post (**c**, **e**) cell injection showing the top view and a depth profile. The opening of the needle is seen on the right side of the images from which the macrophages emerge post injection as a dense line-up (arrows), 0.7–1 mm below the skin surface (–). Blood vessels are faintly detected at 630 nm and indicated by *. Scale bars are 500 μm in x, y, and z. Images at control wavelength 532 nm can be found in Supplemental Fig. 19. Inset in panel (**c**) shows labeled macrophages in histological tissue sections with Schmorl's staining. The outtake corresponds to an area near the needle tip. Scale bar is 50 μm

not suppressing macrophage polarization in response to stimulating factors such as LPS. In contrast, previous studies show the difficulty of labeling pro-inflammatory macrophages with, e.g., NPs, as per cell particle uptake is reduced by ~40%[51]. It has furthermore been demonstrated that metal- or silica NPs can negatively influence cell functions by, e.g., altering expression of extracellular matrix receptors, reorganizing the cytoskeleton or foremost reducing cell migration capability[17,52,53]. In addition, above-mentioned particles trigger innate immune signaling pathways, such as the Nlrp3 inflammasome, leading to the secretion of pro-inflammatory cytokines to exacerbated concurrent diseases, e.g., as shown for IBD[54–56]. While we also experienced reduced in vitro motility of gold nanorods (10 nm diameter Silica-coated)-treated primary macrophages, HDP-labeled cells performed without significant difference to untreated cells.

We demonstrate the functionality of HDP-labeled M1-polarized macrophages in vivo by visualizing their migration to a defined region of a deposited chemical attractant commonly associated with inflammatory events, at 24 h post intravenous injection using whole-animal MSOT imaging in four independent experiments. OA-signal patterning coincides with the distribution of injected labeled cells in animal histology and is a strongly recommended control. Further, HDP signal can be separated from that of hemoglobin using MSOT by demonstrating deep intramuscular injection in a mouse hindlimb. Ultimately, utilizing RSOM, single labeled macrophages are visualized in blood-phantoms in vitro and depicted as a dense line-up of cells released from a catheter needle 1 mm deep in subcutaneous tissue in vivo.

To our knowledge, these data represent the first characterization of HGA-mediated pigment as an OA imaging agent in vitro and in vivo. Our results, especially the in vivo applications, demonstrate the potential of the HDP-marker which now expands the sparse palette of OA labeling agents for immune cells.

In addition, we point out that the labeling process is reversible in macrophages and that HDP-laden cells are detected in vivo up to 48 h post cell injection. It remains to be determined what happens to the label in a healthy animal after that time, but

provided that it is released from the macrophages, it is likely membrane-bound and will presumably be cleared as cellular debris. It can be noted that after systemically receiving HDP-labeled macrophages, the animals did not display signs of distress, nor strong elevation of serum SAA, which would have indicated acute inflammation. In contrast, metal-based nanomaterials accumulate in filtrating organs for a prolonged time, often resulting in local inflammation.

Future efforts will focus on the adaptation of a genetically engineered HDP cell system that can furthermore be expanded to other cell lines or tissues and that will ideally allow expressional regulation to control HDP production. Intracellular production of HDP is conceivable by either depleting homogentisate 1,2-dioxygenase which will elevate intracellular HGA levels or by introducing the key players of HGA synthesis into such cells which lack the enzymatic cascade for tyrosine catabolism.

Together with the associated OA techniques for non-invasive whole-animal imaging, the HDP cell label will provide new insights into the behavior of macrophages during different pathophysiological states by visualizing their spatiotemporal distribution with high resolution in vivo.

## Methods

**Cell culture**. The mouse macrophage lines Ana-1 and J774A.1 (kind gifts of Heiko Adler, HMGU) and the human T lymphocyte line, Jurkat (ATCC), were cultured in RPMI 1640 growth medium, while HEK (human embryonic kidney, ATCC) and HeLa (human cervical cancer, ATCC) lines were grown in Dulbecco's modified Eagle's medium (DMEM), each supplemented with 10% (v/v) fetal calf serum, 100 U/mL Penicillin/Streptomycin solution and 1 mM sodium pyruvate. Cell cultures were split at 70–80% confluency and subsequently transferred into fresh medium or used for experiments. Culture of primary bone marrow-derived macrophages (BMDMs) is described separately. Mouse embryonic stem cells were cultured in DMEM containing 15% (v/v) fetal calf serum, 2 mM glutamine, 0.12 mM non-essential amino acids, 0.1 mM mercaptoethanol and $10^3$ U/ml LIF (leukemia inhibitory factor) at the iPSC (Induced Pluripotent Stem Cell) Core Facility at the Helmholtz Zentrum München— German Research Center for Environmental Health.

**Cell viability**. Viability of a cell culture was assessed by Trypan Blue staining and scoring of dead to viable cells using the Neubauer chamber. Experiments were conducted in duplicates or triplicates. Results were additionally verified using the

automated cell counter Countess II (Invitrogen). In addition, cell viability was determined via LDH (lactate dehydrogenase) assay as described below.

**Preparation of HGA, synthetic melanin, and nanoparticles**. A 100 mM stock solution of homogentisic acid (HGA, Sigma-Aldrich, #H0751) was prepared in cell culture grade PBS, filter-sterilized and single-use aliquots were stored at −80 °C for up to 3 months. For HDP formation in cells, working solutions of 0.1–1.2 mM HGA were tested in cell culture by direct addition to growth medium. As a standard labeling method, we used 0.3 mM HGA for Ana-1 cells and 0.5 mM HGA for BMDMs for a duration of 96 h, unless stated otherwise. Single dose supplementation is sufficient.

The preparation of synthetic (eu-)melanin begins by mixing melanin powder (Sigma-Aldrich, #M8631) with cell culture grade PBS followed by a 3-week incubation period to allow a small fraction of the melanin to solubilize. Working solutions of up to $OD_{700nm}$ 0.1 of the soluble melanin fraction were then applied to cultures of Ana-1 or BMDMs for 24 or 96 h.

Silica-coated gold nanorods with $8 \times 10^{11}$ NPs/ml, a 10 nm diameter and absorption peak at 780 nm were purchased at Sigma-Aldrich (#747971). Macrophages were allowed to phagocytose NPs in the presence of serum-free OptiMEM medium, which improves uptake, and for a duration of up to 24 h. Lower incubation times resulted in poor uptake. NPs were titrated onto sub-confluent cells ranging from 300–30,000 particles per cell assuming maximum uptake. However, we observed that cell uptake remained below 10,000 particles/cell, confirmed by absorption measurements of cell media at start vs end of incubation. Further, we examined coloration of washed cell pellets and OA signal of the NPs-laden cells at 780 nm in MSOT.

**Preparation, differentiation, and activation of macrophages**. Primary bone marrow-derived macrophages (BMDMs) were generated as described earlier[57] by isolation of bone marrow of femur and tibia of FoxN1 nude female mice (Charles River Laboratories, Boston, USA) at 8–10 weeks of age. The following alterations to the protocol were implemented: On day 2 (day 1 = isolation), all adherent cells were discarded by collecting, washing and reseeding only floating cells. On day 5, half of the medium was replaced by freshly prepared medium with the onetime addition of 0.5 mM HGA for HDP formation. Between days 8 and 9, macrophages were gently harvested by accutase-mediated detachment for flow cytometric analysis, in vivo recruitment experiments or for IF and Schmorl's staining (as described below). For cell activation/polarization to pro-inflammatory M1 type, macrophages were treated with 75 ng/ml of bacterial lipopolysaccharide (LPS, Sigma-Aldrich, #L2654) for the last 24 h before cell collection.

**Cell motility assay**. To determine cell motility in the presence of different labels, we deployed a 'scratch' assay which is standard practice[58]. Primary macrophages were prepared as described above and supplemented with either 0.5 mM HGA for 96 h or silica-coated gold nanorods (see above) at $10^{11}$ particles (calculated at ≈30,000 NPs/cell if max. uptake occurs) for 24 h or they remained untreated. LPS (75 ng/ml) was added to each culture for overnight incubation prior to scratch to insure M1 polarization. Each condition (HGA, NPs, WT) was tested in triplicate and resulted in a total area/condition of ≈20 mm² and total cell number/condition of 600–1200 cells being scored 24 h after scratch. Results are presented as averages of cells normalized to area plus standard deviations and a Students $t$-test (tails 2, type 2) was applied to determine significance presented as $p$ values.

**Cytokine/chemokine and LDH release assays**. BMDMs were generated as described above and treated for the last 5 days of differentiation with or without HGA at 0.5 mM for strong HDP pigmentation. Subsequently media was renewed for all samples, accordingly, with or without fresh HGA, and additionally supplemented with or without 200 ng/ml LPS allowing for 3 h of cytokine/chemokine secretion before supernatants were collected. Triplicates were prepared for each condition with $4 \times 10^5$ cells/48-well. Multiplex analysis of secreted cytokines and chemokines was done using the Procarta Plex Mix&Match Mouse (Invitrogen), according to the manufacturers protocol (Invitrogen), and analyzed on a MAGPIX® system (Merck). Cell viability was assessed with the Pierce LDH Cytotoxicity Assay Kit (Thermo Scientific). All results are shown as averages of 3 with standard deviation.

**SAA release assay**. FoxN1 nude female mice aged 8–10 weeks were injected with BMDMs that have been treated with 0.5 mM HGA for 96 h prior to harvest. Cells were PBS-washed three times and cell numbers were determined. In total, $0.6 \times 10^6$ HDP-labeled cells were injected per mouse by tail vein. Steady-state, 24 and 48 h serum levels of SAA were measured using the Mouse Serum Amyloid A DuoSet ELISA (R&D Systems), according to the manufacturers protocol.

**Flow cytometry**. For fluorescence flow cytometric analysis, BMDMs were differentiated up to day 8 after isolation. They were treated in the presence or absence of a single dose of 0.5 mM HGA for days 5–8, as well as with or without 75 ng/ml LPS for the last 24 h to initiate M0 to M1 activation. Cells were gently harvested, washed and stained for 30 min on ice with the following conjugated antibodies diluted 1/100: CD38-FITC (kind gift from Dr. E. Glasmacher), F4/80-APC and

CD11b-FITC (Affymetrix). Flow cytometry was carried out using the BD LSRFortessa (IAF, HMGU). Data analysis was performed with the FlowJo 10 software.

**In vivo recruitment of HDP-labeled cells**. All animal experiments were approved by the government of Upper Bavaria and were carried out in accordance with official guidelines. FoxN1 nude female mice aged 8–10 weeks were utilized for in vivo recruitment experiments. BMDMs were prepared as described above. A single dose of 0.5 mM HGA was added to the growth media on day 5 as well as 75 ng/ml LPS to initiate M0 to M1 activation on day 8. Cells were gently harvested on day 9, washed twice with prewarmed PBS and cell number and viability were determined.

For the injection of BMDMs into the mouse tail vein, prewashed HDP-labeled or unlabeled cells were resuspended in PBS + 2 mM EDTA, filtered through a cell strainer to prevent clumping and immediately injected in a final volume of 200 μl. Prior to cell injection, the recipient animal received two separate subcutaneous matrigel® (Corning, phenol red free, #354262) implantations on the lower dorsal area of the body. Each implant had a volume of 50 μl with only one additionally infused with 200 ng of the recombinant murine cytokine Interferon-γ (IFN-γ, Peprotec, #315-05) as well as 50 ng of LPS to stimulate macrophage recruitment.

For matrigel® + BMDM implantations, a defined number of HDP-labeled or unlabeled cells were directly mixed with matrigel® without IFN-γ or LPS and subcutaneously injected on the lower dorsal area of the animal.

To conduct MSOT imaging, mice were anaesthetized using 2% Isofluran in $O_2$. The anaesthetized mouse was place in the MSOT holder using ultrasound gel and water as coupling media. After completion of the experiments all mice were sacrificed and stored at −80 °C for cryopreservation and subsequent sectioning.

**Preparation of mouse tissue sections**. Tissue sections of cryopreserved mice were prepared using the regions of interest such as the lower abdomen carrying the matrigel® implantations as well as the area of the liver, spleen, and kidneys. We used the Leica CM1950 Cryostat to generate sections of 10-μm thickness. All sections were immediately utilized for staining or further stored at −80 C.

**Immunofluorescence, histochemistry, and microscopy**. For immuno-fluorescence staining of BMDMs pretreated with or without 0.5 mM HGA for a total of 96 h, cells were grown on poly-L-lysine coated coverslips overnight, briefly washed with prewarmed PBS, fixed for 7 min in 4% prewarmed paraformaldehyde (PFA) followed by repeated PBS washes. 0.1% Triton X-100 in PBS was used to permeabilize the cells followed by a block for 30 min with 1% BSA, 10% goat serum and 0.1% Triton X-100 in PBS[59]. The primary polyclonal rabbit antibody to Lamp1 (Lysosome marker, Abcam #24170, 1:1000) was used, followed by the secondary antibody goat anti-rabbit Alexa 488 (ThermoFisher A-11008, 1:500). For the staining of frozen tissue sections, the protocol was altered the following: Sections stored at −80 °C were transferred to room temperature, air dried for 30 min and fixed for 10 min with 4% PFA. The primary rat monoclonal antibody recognizing the mouse F4/80 antigen (Macrophage-specific marker, Abcam #6640, 1:500) was applied, followed by the secondary antibody goat anti-rat Alexa 594 (ThermoFisher A-11007, 1:500). VectaShield plus Dapi mounting medium (Vector Laboratories) was used for all cell staining, while ProLong Diamond antifade mountant (ThermoFisher) was utilized for tissue staining.

For Schmorl's staining, frozen tissue sections were prepared as described above prior to performing the Schmorl's staining method according to the manufacturer's protocol (Morphisto GmbH) with the exception that incubation times were reduced to <1 min. Schmorl's staining solution was prepared immediately prior to use and filtered (0.45 μm) to remove particles. The solution was not used if exposed to oxygen longer than 1 h and contact with metals was prevented by using white plastic forceps for handling of specimens. Schmorl's method uses the reducing properties of melanin and melanin-like pigments such as HDP. Ferricyanide is converted to ferrocyanide which in the presence of ferric ions produces insoluble Prussian blue giving the pigment a blue-green color.

All microscopic images were collected with the Zeiss Axio Imager M2 using the AxioCam MRm or AxioColor cameras and the Plan Apochromat 100×/1.4 oil-, the EC Plan-Neofluar 40×/0.75 Ph2 or −20×/0.5 Ph2 objectives or the Axio Imager Z1 with ApoTome using the 63×/1.2 W Korr UV/VIS objective. Tissue tilings were generated with the Axio Scan.Z1 and a Plan-Apochromat 20×/0.8 M27 objective. The Zeiss Zen software was used for image processing and deconvolution at default (constrained iterative method) was applied to the Lamp1-stained IF images.

Live cell imaging of cells as presented, e.g., in Supplemental Fig. 4 was conducted using the Leica DMI 3000B inverted microscope with C Plan L40×/0.50 PH2 and L20×/0.40 objectives. Images were acquired using the Leica application suite software.

**Raster-scan optoacoustic mesoscopy (RSOM)**. For RSOM measurements we employed a systems as described in ref. [60]. Briefly, a broadband spherically focused transducer with central frequency of 54.2 MHz was used to record Optoacoustic signals generated in the sample by 6 ns pulses emitted by a diode-pumped Nd:YAG optical parametric oscillator (OPO) with a pulse repletion rate of 50 Hz (SpitLight OPO, InnoLas Laser GmbH, Krailling, Germany). After amplifying using a 63 dB

low noise amplifier (Miteq AU-1291, USA), data was recorded by a high-speed digitizer operated at a sampling rate of 500 MHz (CS122G1, Gage, USA). Each phantom (blood-agar + HDP-labeled macrophages as well as pure blood-agar) was imaged with a step size of 20 μm and a scanning area of 4 mm × 3 mm at 630 nm. The datasets are frequency filtered and reconstructed using a tomographic algorithm, yielding an image for a frequency range between 10 and 70 MHz displayed in a gray scale. Reconstructed images were filtered with a 3-by-3 neighborhood median filter and the intensity was balanced with a power function of 0.43 to equalize the needle signal and avoid image saturation. The volumetric scatter plot from the cells embedded in the blood-agar droplet was obtained from the 3D-binarization of the OA reconstructed volume by applying a global threshold of 0.51 using the imbinarize function in Matlab (R2017b). A clustering analysis of the binary data was then performed using the regionprops3 function in Matlab to obtain the size distribution of clustered volumes and compute the histogram of cell diameters. For in vivo visualization of cells, a catheter was subcutaneously placed on the lower back of a FoxN1 nude mouse after the animal was anaesthetized using 2% Isofluran in $O_2$ and placed on a 33 °C warming pad. The scan head was positioned on the skin above the tip of the catheter needle scanning a region of 3 mm × 2 mm. A 'pre-scan' was conducted after minimal injection of PBS (to prevent air injection) followed by the injection of HDP-labeled primary macrophages and an immediate 'post-scan'. We cannot exclude the possibility of cell motion during imaging. The procedure is outlined in (Supplemental Figs. 17 and 18).

**OA microscopy**. Optoacoustic microscopy images were recorded using 592 nm laser light excitation (Katana 06-HP, OneFive, Switzerland) and a 20 MHz single-element transducer in trans-illumination configuration for detection of generated acoustic waves (Imasonic, France). The laser light was focused down to a 250 nm spot using an Olympus UPLSAPO 63XW 1.2 NA objective. Electrical signals from the transducer were amplified by a 63 dB low noise amplifier (Miteq AU-1291, USA) and digitized using 3 GS/s DAQ card (GaGe Applied EON Express, USA). Imaging parameters: scanning step size 200 nm, exposure time 2.5 ms, laser power 20 mW.

**MSOT data acquisition, reconstruction, and unmixing**. All MSOT data has been recorded using the MSOT inVision 256 (iThera Medical GmbH, Munich, Germany).

In vitro: MSOT spectra between 680 and 900 nm of HDP-labeled macrophage preparations have been recorded using clear plastic syringes together with an equally prepared black indian ink reference with $OD_{700nm}$ at 0.3. Cell number per syringe ($8 \times 10^6$) is kept constant throughout all experiments. A range of copper sulfate concentrations has been used to correct for day-to-day intensity variations. Using Matlab (R2017b) images have been reconstructed using a model based algorithm[61] and filtered for the frequency range between 0.3 and 7 MHz. Speed of sound was regularly set to 1530 m/s and a regularization parameter was used with $10^6$. The field-of-view (FOV) was $250 \times 250$ pixel with a pixel resolution of $200 \times 200$ μm. Details of MSOT recording are described elsewhere[62]. Mean MSOT intensities for each sample were directly extracted from six different regions of interest (ROI) covering the syringe diameter. Experiments were prepared in duplicates or triplicates.

In vivo: Mouse MSOT data between 680 and 960 nm has been recorded as described above with 0.3 mm spacing covering the region of the matrigel® implant as well as the regions of liver and spleen. Due to their lack of spectral features melanin-like spectra are notoriously hard to unmix. Due to the imaging of recruited immune cells we observe relatively low cell numbers dispersed throughout an area which results in a comparably low label density. In brief (Supplemental Fig. 16): (i) In Matlab (R2018), spectra of all pixels within the slices of interest within the boundaries of the mouse-body-ROI were clustered using a k-means clustering algorithm with a fixed bin number of bins (200, different bin numbers have been tested but did not achieved better results). (ii) Based on experimental knowledge (here bait-implant and control-implant) the data was split in the mouse-body-ROI along the median plane and selected for clusters that had a predominant occurrence in one of the two areas (80%> evenly distributed). (iii) From this set of clusters those that exhibit a high similarity to the most melanin-like spectral cluster were selected (covariance < 0.98). (iv) The remaining clusters were binned and visualized after one cycle of pixel homogenization as sum projection on the anatomy images at 720 nm as average projection. For the corresponding spectra the mean has been calculated after normalization and shown along with the image data. We argue that the analysis strategy is especially suitable for low and potentially disperse signals. It can be adapted to other questions by subdividing the imaging data and comparing the regions based on spectral clusters exploiting the full multispectral capacity of MSOT. Manual inspection of spectral clusters with prevalence in specific ROIs secures identification of even small spectral differences reminiscent of the presence of labels in this area.

**UV/VIS spectroscopy of HDP formation**. To determine the absorption spectra of HDP formation in RPMI media in the absence or presence of Ana-1 cells or Ana-1 cellular debris (sonication), samples were taken at 0, 24, 48, 72, and 96 h after addition of 0.3 mM HGA in media. Samples were spun down for 5 min at 2000 rpm to pellet cells or for 5 min at 14,000 rpm to pellet cell debris and the resulting supernatant was diluted 1:10 prior to measurements using the Shimadzu UV-1800 Spectrometer.

**Reporting summary**. Further information on research design is available in the Nature Research Reporting Summary linked to this article.

## Data availability
A reduced source data contents file is available online including the data organized by figure numbers in the paper. Due to size limitations of depositing raw imaging data the data that support the findings of this study are available from the corresponding author upon request.

## Code availability
Code is available from the corresponding author upon request.

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

## Acknowledgements

The authors wish to thank Ruth Hillermann and Sarah Glasl for technical assistance as well as Benjamin Schnautz for assistance with FACS, Anna Pertek and Micha Drukker for providing mouse ES cells, Andrei Berezhnoi for technical assistance with RSOM, the Institute for Pathology at Helmholtz Zentrum München for providing the Axio Scan.Z1, and Armando C. Rodríguez for discussions on the paper. C.Z. has received funding from the European Union's Horizon 2020 research and innovation program under grant agreement No. 732720 (ESOTRAC). P.D. was funded by the German Research Foundation (DFG) (SFB1123 and DU1522 1-1). J.A. received funding from European Grant INNODERM (687866) Horizon 2020. A.C.S. receives funding from DFG (STI656/1-1).

## Author contributions

I.W. and A.C.S. conceived experiments and wrote the paper. I.W. conducted all experimental work, with contributions from U.K., and analyzed the data except the following: J.A. conceptualized and demonstrated single-cell imaging in RSOM and C.Z. performed and analyzed single-cell RSOM measurements, P.D. performed the Multiplex and ELISA analyses, A.C. conducted OA Microscopy, and A.C.S. analyzed the mouse MSOT data. V.N. contributed to the paper.

## Competing interests

V.N. is a shareholder of iThera Medical GmbH. All other authors declare no competing interests.
