## [Peer Review File · Nature Communications]

Reviewers' comments:

Reviewer #1 (Remarks to the Author):

The authors propose an HGA-derived pigment for macrophage labeling finalized to optoacoustic in vivo imaging. Optoacoustic imaging emerged as a new biomedical imaging technology and currently efforts are being made towards non-toxic biodegradable contrast agents. Endogenous molecules, such as hemoglobin of the blood and melanin, are the main tissue chromophores that can generate a strong optoacoustic signal, although they can be applied only in limited biological investigations with sub-optimal contrast capability. Therefore, the design of exogenous contrast agents may play a key role in improving contrast and imaging accuracy, and in providing specific molecular information. In this work, authors discuss the physicochemical characterization and optoacoustic imaging applications of an HGA-derived pigment capable of controlled accumulation in the target cells with sub-optimal contrast capability.

This paper may potentially add new knowledge to the growing research into this recent field of imaging, but it needs to be deeply revised since different crucial points are to be addressed. Some of the experimental results are questionable and it would also be beneficial to carry out complementary experiments to support the reported findings. Detailed comments are listed here below.

First of all, HGA is the etiologic agent of an inborn error of metabolism, the genetic disease alkaptonuria, an invalidating multi-systemic pathology. Although the authors mention the disease in the introduction, they seem not to be aware about the increasing literature on alkaptonuria and HGA-related tissue and organ damages. The pathophysiological role of HGA must be well described, although still to be fully determined or maybe just in virtue of many still obscure molecular mechanisms in which HGA is certainly involved, starting from its unknown auto-polymerization process. It is today well known that HGA should be completely absent in a healthy human organism as well as many reports have been published on the cytotoxic disrupting devastating effects on human alkaptonuric cells and tissues; thus, all the most recent relevant papers should be cited by the authors.

Authors reported the use of HGA in primary and immortalized macrophages. Literature on AKU is unanimous on the concept that HGA is differently toxic if exogenously administered to cell lines or primary cells. Many recent works report unequivocally ascertained pro-oxidant and pro-inflammatory actions of HGA on human cell models based on primary cells. Authors must be aware of these fundamental data when they suggest the use of HGA as an injectable molecule for optoacoustic studies in vivo. In virtue of this, authors must substantiate their assertions with more detailed experiments on viability of primary cells supplemented with HGA. Data reported in the Supplementary material are not sufficient to ensure that HGA is a safe molecule for this kind of purpose.

Authors state also that "HDP label is polymerized in living cells, thereby greatly reducing stress and cytotoxicity". This is a quite disputable statement on the basis of what so far reported on HGA. In alkaptonuric patients, HGA tissue accumulation leads to severe inflammation, tissue degradation and oxidative stress, as reported in many papers (that authors should cite), clearly indicating that HGA is toxic. Moreover, published studies (to be cited) on the ochronotic HGA-derived pigment confirmed its insoluble nature and the resistance to a plethora of biological and chemical cleavage agents (thus making HGA-derived pigment non-biodegradable). In addition, alkaptonuria has been indicated to be a novel amyloidogenic disease and in fact, HGA was reported to be a very strong promoter and enhancer of amyloid aggregation and fibrillation (Braconi et al 2017) for different kinds of proteins and peptides, even at extremely low concentrations (far much lower than HGA blood levels in alkaptonuric patients), inducing and accelerating the production of melanin-like pigmented toxic insoluble aggregates. The authors never mentioned this peculiar property of HGA that could have a very dangerous effect once injected in a living organism. HGA pigment solubility is a key factor in determining optoacoustic efficiency, the capability to control its aggregation state is essential for future developments within the field of optoacoustic imaging and sensing. How can authors justify the perfect solubility of the HGA-derived pigment they used and the discrepancy with published data? How can authors control an aggregation process that is still largely uninvestigated? Authors refer in their manuscript to 10-20 kDa HGA-oligomers, citing a paper on bacterial melanin (pyomelanin), that is

different from human melanin (eumelanin) and again different from the melanin-like HGA-derived ochronotic pigment in human alkaptonuria. Such confusing information are present all throughout the manuscript and may be misleading for the reader.

Moreover, it is crucial to note that, following intravenous administration, the HGA polymer may present a prolonged retention time in the blood with consequent uptake especially in the spleen and in the kidney. This may be very dangerous and cannot be overlooked in order to make HGA competitive for biomedical imaging applications as an optoacoustic contrast agent. Some concerns about the possible long-term toxicity following the free HGA release needs absolutely to be investigated. Further experiments are needed to assess any toxic effect following accumulation in other organs exposed to HGA, such as heart, liver and kidneys, where aggregated particles may remain for long times.

Authors state also that "The colorimetric MTT assay was impractical due to signal interference from the HDP". Such an assertion is totally in contrast with published reports, since almost all papers (easy to find in commonly used databases) reporting cytological tests following HGA administration to cells indicated MTT as the standard assay to analyze cell viability and metabolic activity of cell populations. Once again, the authors seem to be not sufficiently aware of the big amount of literature existing on the field and should at least cite those papers and explain why they found an interference in MTT assay differently from other previous authors.

Again, authors state that "HDP is similar to pyomelanin, which in turn shows a spectrum similar to that of eumelanin". Apart from the spectrum, this sentence is quite questionable due to the multifaceted nature of melanin. Melanin is frequently considered just an animal cutaneous pigment and is treated separately from similar fungal or bacterial pigments. Pyomelanin production in microorganisms often is associated with numerous survival advantages and was first characterized in bacteria. In pathogenic microorganisms, melanization becomes a virulence factor since melanin protects microbial cells from defense mechanisms in the infected host. These pigments sometimes behave as a double-edged sword and such a double edge should be taken into account in studies for melanin-related treatments. For these reasons, it very hazardous to arbitrarily equal HGA-derived polymers and pyomelanin as well as to compare pyomelanin to human melanin. If authors based their conclusion on such an assertion, they must deeply revise the presented use of HGA as a potential but harmless melanin replacement.

Another crucial point is the protocol of solubilization of HGA. Authors used PBS to dissolve HGA, but this medium may influence the aggregation properties of the molecule. The pH-induced aggregation of melanin particles for monitoring tumor acidic microenvironment is well-known and in virtue of this the authors should examine the efficacy of HGA polymer for specific in vivo tumor imaging. In fact, HGA-laden macrophages can mimic melanoma cells and this needs to be considered in the differential diagnosis.

Moreover, to verify the macrophagic functionality after HDP-labeling, authors used a CD-38 based flow cytometry assay (page 10), but, following the literature, a commonly accepted marker profile for M1-macrophages is CD68+/CD80+, whereas M2-macrophages are characterized as CD68+/CD163+ (Khrantsova G et al 2009 and Gordon S 2003). Can authors justify their choice? Can they provide these further functional assays?

A final but relevant consideration: the authors propose here a highly controversial molecule to avoid the limits of natural and synthetic melanins, but non-specific melanin-based probes for optoacoustic imaging have been just recently synthesized (Liopo et al 2015, Longo et al 2017, Repenko et al 2015), with an excellent optoacoustic contrast and a detection limit of such linear melanin about two orders of magnitude higher than that of the gold nanoparticles. In virtue of this, the use of HGA derived polymer seems not to be a good choice. And again, none of these papers has been cited by the authors in order to compare different probes.

Concluding, there is a significant need for standardized procedures for characterizing this HGA-derived polymer, as well as for protocols for its use in animal models (mice, as previously reported, have an endogenous production of ascorbic acid contrasting ochronotic HGA-based pigment production and are not the best animal model to evaluate HGA-induced pigmentation), in order to improve multi-site comparison. Furthermore, to date, no standard protocols have been established to assess long term stability, accumulation, biodistribution, and safety that are mandatory to promote translation between basic research and clinical applications.

Reviewer #2 (Remarks to the Author):

A. Summary of the key results as interpreted by the reviewer

The present study is the first to demonstrate the OA properties of HDP in vitro and in vivo and to perform pilot studies of HDP application for intracellular labeling of macrophages.

B. Originality and interest: if not novel, please give references

The manuscript describes the original research. The idea of using homogentisic acid-derived pigment as a biocompatible nanoparticle to enhance contrast of optoacoustic imaging is novel.

C. Data & methodology: validity of approach, quality of data, quality of presentation

Methodology is HDP production is well described, with appropriate controls. However, the methodology of HDP characterization is weak. Even though authors state that the optoacoustic contrast is strong, there is no quantitative numbers of such contrast in terms of minimally detectable concentration, increase of optoacoustic contrast per unit volume of HDP, absolute amplitude of optoacoustic signal as a function of concentration.

D. Appropriate use of statistics and treatment of uncertainties

This is an exploratory study, so statistically significant data are not presented. Authors presented only demonstration of feasibility using single experiments.

E. Conclusions: robustness, validity, reliability

This manuscript describes a pilot research studies. Therefore, conclusions are somewhat ostensible.

F. Suggested improvements: experiments, data for possible revision

The results obtained in vitro should provide more details in terms of optoacoustic signal contrast. Concentrations in the range of mM seem too high for physiologically relevant conditions. Optoacoustic signal amplitude is provided on a relative scale and no comparison is provided with background absorption of blood. Authors need to provide absolute values and quantities relevant to optoacoustic images, such as optical absorption coefficient associated with specific concentration of HDP. Also authors obtained images of raster scan optoacoustic microscopy showing existence of optoacoustic contrast in cells, but in vivo optoacoustic imaging data is not convincing. Authors used MSOT system to image small portions of a mouse body, but did not show detectable accumulation of their HDP contrast agent... It seems like the contrast was not so great, so 3D images were not presented...

G. References: appropriate credit to previous work?

The area of optoacoustic imaging research using various nanoparticles, including biodegradable nanoparticles, has been of significant interest for almost two decades. However, authors refer mostly to their own works. Authors should search Scopus and other sources and provide relevant references that created basis of their own research and described significant accomplishments in the field nanoparticle enhanced optoacoustic imaging. As an example, this manuscript compares HDP with Eumelanin. However, they do not refer neither to the originally published photoacoustic images of genetically induced production of eumelanin nor melanin nanoparticles.

H. Clarity and context: lucidity of abstract/summary, appropriateness of abstract, introduction and conclusions

The abstract is formulated stronger than the results obtained. The conclusions need to be revised to state what was truly accomplished and remove forward looking statements. For example, detection of HDP on the background blood in the whole live animal was not demonstrated.

I. Minor edits

Replace "strong contrast in optoacoustic" with "strong optoacoustic contrast" in the Abstract.

Reviewer #3 (Remarks to the Author):

The manuscript by Stiel and colleagues have reported a new approach for labeling macrophages using HDP as optoacoustic imaging contrast. This timely work is significant because labeling macrophages without altering their viability and functions have been a technical challenge. The capability of labeling macrophages using HDP for in vivo tracking of the immune responses will have broad impact for a wide range of fundamental studies, especially for cancer research. The combination of HDP, which is water soluble, with optoacoustic imaging is also novel because it will potentially enable whole-body small animal imaging of the macrophage migration, with high-resolution and high-sensitivity. The paper was well written and easy to follow with sufficient details. With that, I feel that the manuscript is exciting but premature for publication in Nature Comm. My major and minor concerns are listed below.

1. The first major concern is the lack of chemical and biological studies about HDP, which in some cases is a pathological byproduct in alkaptonuria. It raises concerns about the biosafety of using this pigment as the macrophage labeling without a thorough and rigorous investigation about the pigment itself. I agree that the manuscript has presented a large number of experiments showing that HDP-labeled macrophages have apparently normal morphology and functions, which, however, still falls short to make up the lack of knowledge about the HDP itself. This actually suggests to me that this manuscript is better to be combined with more fundamental studies about the chemical, optical (physical) and biological properties of the HDP.
2. The second concern is about the advantage of using of HDP as a marker for optoacoustic imaging in vivo. The authors in the introduction have pointed out that the cell viability is 'greatly affected by random intracellular enrichment of eumelanin'. This statement is not consistent with the published results from the UCL group using tyrosine (Nature Photonics volume 9, pages 239–246 (2015)), in which the heavy expression of melanin does not seem to reduce the cell growth rates of multiple cell lines. While this might be a different case for macrophages, the authors need more evidence on this statement because genetically encoded contrast is clearly superior over the ex vivo labeling contrast. Additional experiments using eumelanin-expressing macrophages are needed for a fair comparison.
3. Following the comment above, despite the good performance of HDP-labeling, its application may be greatly limited by the fact that ex vivo labeling of the macrophages is needed before these cells can be returned to the animals to participate in the normal immune process. I agree that the present work is already a step forward for imaging macrophages in action. The need for ex vivo labeling and cell culturing have moderately reduced its potential applications and thus impact.
4. It is clear that high concentration HGA (more than 1 mM) is harmful to the cells. In this case, it is critically important to analyze the underlying mechanisms of HGA or HDP's cell-toxicity. A viable cell is not necessarily functioning normally.
5. Minor. In the plots showing the optoacoustic signals, 'intensity' should be 'amplitude' because it is acoustic signals.
6. The single-cell visualization in Fig. 4c is not very convincing. First of all, the RSOM system should not be able to resolve single cells with a 50 MHz transducer if the light is not tightly focused. How was the single-cell visualization validated? Further, the unmixing of the signals into red (low frequency) and green (high frequency) channels is not clear to me. Why should the macrophage signals be high frequencies? What is the cut-off frequency between the red and green channels?
7. Minor. The OA-microscopy is using a 20 MHz transducer but a 3 GHz DAQ card? Is this an overkill for the sampling frequency? What does 'exposure 2.5 ms' mean for OA-microscopy?
8. Figure 4A. The multi-spectral imaging by MSOT is interesting. The extracted HDP signals were extremely sharp in the images. Given that the spectral coloring effect is always an issue for optoacoustic imaging which may induce errors in spectral unmixing, what was exactly the method to extract the HDP signals, and how were the imaging results validated?
9. Minor. Please add colorbars to the figures when applicable.

Reviewer #1

We thank the reviewer for the dedicated, insightful and highly educated feedback given to us. We realize that the reviewer feels very strongly about the research on alkaptonuria and the effects of ochronosis in patients and we greatly respect all endeavors of this research field. In the following chapter, it is our aim to address questions and concerns that may still exist and to clarify, once more, how our system differs from alkaptonuria and how we have turned HGA-derived pigment into a safe and efficient label for OA macrophage imaging in vivo.

The authors propose an HGA-derived pigment for macrophage labeling finalized to optoacoustic in vivo imaging. Optoacoustic imaging emerged as a new biomedical imaging technology and currently efforts are being made towards non-toxic biodegradable contrast agents. Endogenous molecules, such as hemoglobin of the blood and melanin, are the main tissue chromophores that can generate a strong optoacoustic signal, although they can be applied only in limited biological investigations with sub-optimal contrast capability. Therefore, the design of exogenous contrast agents may play a key role in improving contrast and imaging accuracy, and in providing specific molecular information. In this work, authors discuss the physicochemical characterization and optoacoustic imaging applications of an HGA derived pigment capable of controlled accumulation in the target cells with sub-optimal contrast capability. This paper may potentially add new knowledge to the growing research into this recent field of imaging, but it needs to be deeply revised since different crucial points are to be addressed. Some of the experimental results are questionable and it would also be beneficial to carry out complementary experiments to support the reported findings. Detailed comments are listed here below.

First of all, HGA is the etiologic agent of an inborn error of metabolism, the genetic disease alkaptonuria, an invalidating multi-systemic pathology. Although the authors mention the disease in the introduction, they seem not to be aware about the increasing literature on alkaptonuria and HGA-related tissue and organ damages. The pathophysiological role of HGA must be well described, although still to be fully determined or maybe just in virtue of many still obscure molecular mechanisms in which HGA is certainly involved, starting from its unknown auto-polymerization process. It is today well known that HGA should be completely absent in a healthy human organism as well as many reports have been published on the cytotoxic disrupting devastating effects on human alkaptonuric cells and tissues; thus, all the most recent relevant papers should be cited by the authors.

Authors reported the use of HGA in primary and immortalized macrophages. Literature on AKU is unanimous on the concept that HGA is differently toxic if exogenously administered to cell lines or primary cells. Many recent works report unequivocally ascertained pro-oxidant and pro-inflammatory actions of HGA on human cell models based on primary cells. Authors must be aware of these fundamental data when they suggest the use of HGA as an injectable molecule for optoacoustic studies in vivo. In virtue of this, authors must substantiate their assertions with more detailed experiments on viability of primary cells supplemented with HGA. Data reported in the Supplementary material are not sufficient to ensure that HGA is a safe molecule for this kind of purpose.

In this study, we set out to develop an OA-labeling method for macrophages. After testing different concentrations of HGA, we chose a working concentration of 0.3 mM for immortalized and 0.5 mM for primary macrophages which, in both cases, ensures unperturbed cell viability, comparable to that of untreated cells. HGA only begins to decrease cell viability of macrophages at very high concentrations, which we discuss early on in the results and show in Figure 1. While we routinely scored cell viability using Trypan Blue staining, we now added a widely used LDH assay (lactate dehydrogenase; Suppl. Fig S7) confirming our previous results. It has to be strongly emphasized at this point, that we do not ‘*use HGA as an injectable molecule...in vivo*’, but rather label the macrophages *in vitro* by HGA treatment (4-5 days) and subsequently inject the cells with the intracellular pigment into the mouse.

Concerning the genetic disorder, alkaptonuria, we thank the reviewer for suggesting to strengthen the discussion of the pathophysiological aspects of HGA by adding a designated paragraph about alkaptonuria in our introduction (page 3, 1st paragraph). Several paper references were added to inform the reader about the current disease-related research. We are aware, that patients of advanced age show formation of black pigmented plaques - ochronotic pigment - preferentially located on collagen fibers of connective tissues of cartilage and bone (Tinti, 2011; Laschi, 2012), and secrete increased proinflammatory cytokines, as well as SAA, commonly observed during a chronic disease (Millucci, 2012). Further, healthy chondrocytes

treated with 0.3 mM HGA for a prolonged time (8 days) can reproduce the above-mentioned phenotypes with a noticeable overexpression of IL-6, various other cytokines, and even SAA (Spreafico, 2013). Nonetheless, there seems to be a distinct dependency on cell type. In our work, we extensively studied the wellbeing of primary – and immortalized macrophages in the presence of increasing concentrations of HGA and over the duration of several days. For this resubmission we were motivated by Spreafico et al. to add a multiplex assay, where secretion of 24 different cytokines/chemokines of macrophages were measured in the presence and absence of HGA (0.5 mM for 96h) and +/- LPS (representatives shown in Suppl. Figure S7). We can now show that HGA-derived pigment has no detectable intrinsic activity in primary macrophages, nor does it induce specific M1- or M2-like polarization. HGA is non-toxic for macrophages (see LDH release assay), nor does it induce an inflammatory response in these cells. In addition, HGA does not perturb macrophage functionality: HDP-labeled cells can still be activated after treatment with LPS as demonstrated by Multiplex and FACS (Suppl. Figures S7 and S6). Moreover, analysis of cell motility in the presence and absence of pigment showed no significant difference, in contrast to treatment with SiAu nanorods, which greatly reduced cell motility of primary macrophages (Figure 1g) and which we chose as a comparative OA agent for resubmission.

Normal expression levels of IL-1 and IL-6 in HDP-cells ensures that there is no acute phase protein reaction caused by HGA or its intracellular pigment. Consequently, it is safe to exclude any formation of SAA-based amyloidosis in our cell system. Further, serum of animals (n=3) receiving HDP-labeled macrophages showed SAA levels < 10 µg/ml and only a small increase in comparison to base line. This is most likely due to the tail vein injection procedure itself and expected for small interventions in FoxN1 mice (Noguchi-Sasaki, 2016). We added the respective data on page 8, 3rd paragraph.

Furthermore, we find no evidence of intercellular transfer of pigment from labeled macrophages to other cells *in vitro* (Suppl. Figure S9) nor of significant release of the pigment into supernatants after pigmented cells are transferred to fresh media (Suppl. Figure S4). Together, this suggests, that no significant amount of free HGA or pigment reaches the organism via the labeled macrophages, nor does the presence of HDP-labeled macrophages *in vivo* induce inflammation in test animals. However, if HGA should be released by the labeled cells *in vivo*, the presence of HGD enzyme in the healthy animals would lead to its degradation. To our best knowledge, we exclude any health hazards for these mice. Lastly, we also do not advertise this label to be used for clinical applications, we emphasize that it will facilitate basic research focusing on non-invasive *in vivo* studies of macrophage distribution and functionality in health and disease.

Authors state also that “HDP label is polymerized in living cells, thereby greatly reducing stress and cytotoxicity”. This is a quite disputable statement on the basis of what so far reported on HGA. In alkaptonuric patients, HGA tissue accumulation leads to severe inflammation, tissue degradation and oxidative stress, as reported in many papers (that authors should cite), clearly indicating that HGA is toxic. Moreover, published studies (to be cited) on the ochronotic HGA-derived pigment confirmed its insoluble nature and the resistance to a plethora of biological and chemical cleavage agents (thus making HGA-derived pigment non-biodegradable).

In the results section ‘Cellular pathways for HDP formation in macrophages’ we discuss the possible scenario leading to intracellular pigmentation of macrophages via HGA (model in Figure 1d). We reference existing studies and conduct our own, listed in that corresponding chapter, from which we conclude that macrophages prefer to take up HGA and small BQA intermediaries (auto-oxidation product of HGA) after which HDP - generating strong OA signal at <600nm wavelength - gradually forms within the cell. It is conceivable that cellular materials, e.g. amino acids, are incorporated during polymerization. In the manuscript we argue that although the exact structure and composition of the present HGA-derived pigment in macrophages remains to be determined, it is a soluble melanin species and therefore related to, not identical with, pyomelanin. We cite studies demonstrating that both bacterial pyomelanin and pure HGA-pigment are small polymers of 10-20 kDa (Roberts, 2015). We see that HDP forms in aqueous solution and centrifugation at 14000 rpm for 2h does not produce a sediment (data not shown).

In macrophages, we find HDP localized within Lamp1-positive vesicles which identify as endolysosomes (Figure 2). It is conceivable that this compartmentalization, which is also seen for eumelanin in melanosomes, is partially responsible for the compatibility of macrophages and HDP. We want to emphasize a likely difference to ochronotic pigment, which requires decades of continuous HGA buildup in patients' serum, urine, and interstitial fluids to result in the characteristic ochronotic pigment deposits primarily found on collagen fibers of connective tissues of cartilage and bone. This indicates that ochronotic pigment is preferentially formed in the extracellular spaces, although cell models, osteosarcoma and chondrocytes (e.g. Tinti 2011), with intracellular deposits have been published.

In our study, HDP is formed in a defined system containing RPMI media, HGA and macrophages. Although we cannot exclude complexation with amino acids in our scenario, the pigment produced in cultures of macrophages is not identical to the ochronotic pigment of alkaptonuria patients. To emphasize this difference we revised the results section "*Cellular pathways for HDP formation in macrophages*".

We refer to bio-compatibility in comparison to agents such as e.g. metal nanoparticles. To this end, we added a comparison between HGA-labeled - and nanoparticle-fed cells (10 nm SiAu nanorods) (Figure 1 e-g). Although comparable in OA signal generation and cell viability, HDP label suggests preservation of macrophage functionality, compared to SiAu nanorods, as shown in motility assays (Figure 1 g). As HDP also does not alter cytokine release in labeled macrophages (see above), nor inhibit LPS-induced activation of these cells, or alter cell viability, this too must be considered as bio-compatibility.

In addition, alkaptonuria has been indicated to be a novel amyloidogenic disease and in fact, HGA was reported to be a very strong promoter and enhancer of amyloid aggregation and fibrillation (Braconi et al 2017) for different kinds of proteins and peptides, even at extremely low concentrations (far much lower than HGA blood levels in alkaptonuric patients), inducing and accelerating the production of melanin-like pigmented toxic insoluble aggregates. The authors never mentioned this peculiar property of HGA that could have a very dangerous effect once injected in a living organism.

We thank the reviewer for making us aware of this. Amyloidosis is a hallmark of chronic inflammatory diseases and alkaptonuria patients tested positive for SAA and SAP, e.g. in the paper of Mellucci 2012, were between 45-69 years of age. There seems to be a correlation between progression of disease and SAA amyloid – pigment coaggregations. Interestingly, *in vitro* cell cultures of chondrocytes treated with HGA will also show elevated SAA levels as publishes by others. Our primary macrophages grown in the presence and absence of 0.5 mM HGA for 5 days had normal levels of IL-6 (Suppl. Figure S7). This cytokine must be elevated in order to enhance the production of SAA. To be safe, we directly measured SAA in above macrophages which had no detectable levels (page 8, 3rd paragraph).

In summary, for our labeling purpose we can only discuss our time-window of observation (< 5d). In this time HGA or the derived pigment does not alter the functionality of macrophages making a strong phenotypical change of the cell due to oxidative stress of fibrillation occurs.

HGA pigment solubility is a key factor in determining optoacoustic efficiency the capability to control its aggregation state is essential for future developments within the field of optoacoustic imaging and sensing.

How can authors justify the perfect solubility of the HGA-derived pigment they used and the discrepancy with published data?

We thank the reviewer for prompting us to be more precise on the HGA-derived pigment described in this study and please see above for discussions on that topic. In brief, the polymerization of HGA-derived pigment occurs *in vitro* in the presence of primary macrophages and RPMI media. We do not claim any resemblance to either ochronotic pigment derived from patients nor pigment derived from *in vitro* polymerization of HGA in the presence of healthy or patient chondrocytes.

How can authors control an aggregation process that is still largely uninvestigated?

We do not attempt to fully control the polymerization process which is a highly active research field of its own, however, we monitor the OA-signal generation as a function of HGA concentration and exposure time while considering cell-viability, inflammatory cytokine secretion, macrophage polarization and functionality. With this we ascertain that HDP labeled macrophages can be used to follow macrophage mobility *in vivo* without perturbations. It must be emphasized again, that we do not inject HGA systemically into an animal.

Authors refer in their manuscript to 10-20 kDa HGA-oligomers, citing a paper on bacterial melanin (pyomelanin), that is different from human melanin (eumelanin) and again different from the melanin-like HGA-derived ochronotic pigment in human alkaptonuria.

Please see above for our comments on the nature of HDP. However, to further clarify this in the manuscript, we reference a study which compares pigment sizes of pyomelanin and pure HGA-pigment (page 3, 2nd paragraph). We also compare pyomelanin, eumelanin and the present HDP in terms of their absorption spectra which are also very similar.

Such confusing information are present all throughout the manuscript and may be misleading for the reader. Moreover, it is crucial to note that, following intravenous administration, the HGA polymer may present a prolonged retention time in the blood with consequent uptake especially in the spleen and in the kidney. This may be very dangerous and cannot be overlooked in order to make HGA competitive for biomedical imaging applications as an optoacoustic contrast agent. Some concerns about the possible long-term toxicity following the free HGA release needs absolutely to be investigated. Further experiments are needed to assess any toxic effect following accumulation in other organs exposed to HGA, such as heart, liver and kidneys, where aggregated particles may remain for long times.

We hope that by adding above paragraphs we established clarity concerning the distinction of HDP from other pigments of the same class.

Authors state also that “The colorimetric MTT assay was impractical due to signal interference from the HDP”. Such an assertion is totally in contrast with published reports, since almost all papers (easy to find in commonly used databases) reporting cytological tests following HGA administration to cells indicated MTT as the standard assay to analyze cell viability and metabolic activity of cell populations. Once again, the authors seem to be not sufficiently aware of the big amount of literature existing on the field and should at least cite those papers and explain why they found an interference in MTT assay differently from other previous authors.

We choose to perform trypan blue staining and manual scoring of viable/dead ratios to prevent any interference of the HDP pigment in automated absorbance-based assays, as described for aromatic polymers (Moreno-Villoslada 2007 and 2008). Furthermore, we used LDH levels in the supernatant of cells media as an indicator for cell viability. Since we did not test a potential interference we rephrased page 4, 2nd paragraph to reflect this.

Again, authors state that “HDP is similar to pyomelanin, which in turn shows a spectrum similar to that of eumelanin”. Apart from the spectrum, this sentence is quite questionable due to the multifaceted nature of melanin. Melanin is frequently considered just an animal cutaneous pigment and is treated separately from similar fungal or bacterial pigments. Pyomelanin production in microorganisms often is associated with numerous survival advantages and was first characterized in bacteria. In pathogenic microorganisms, melanization becomes a virulence factor since melanin protects microbial cells from defense mechanisms in the infected host. These pigments sometimes behave as a double-edged sword and such a double edge should be taken into account in studies for melanin-related treatments. For these reasons, it very hazardous to arbitrarily equal HGA-derived polymers and pyomelanin as well as to compare pyomelanin to human melanin. If authors based their conclusion on such an assertion, they

must deeply revise the presented use of HGA as a potential but harmless melanin replacement.

Please see previous answers above. We compare the pigments merely on the spectral properties and draw the comparison to pyomelanin based on the common precursor HGA (Roberts 2015). We are well aware of the complex nature of the melanin family.

Another crucial point is the protocol of solubilization of HGA. Authors used PBS to dissolve HGA, but this medium may influence the aggregation properties of the molecule. The pH-induced aggregation of melanin particles for monitoring tumor acidic microenvironment is well-known and in virtue of this the authors should examine the efficacy of HGA polymer for specific *in vivo* tumor imaging. In fact, HGA-laden macrophages can mimic melanoma cells and this needs to be considered in the differential diagnosis.

Cell culture grade PBS is of neutral pH and ideally suited to dissolve homogentisic acid. It furthermore helps to maintain a constant pH value.

We are aware of the similar OA spectra of melanoma cells, however we do not advertise HDP-labeled macrophages to study melanoma or any other tumor microenvironment.

Moreover, to verify the macrophagic functionality after HDP-labeling, authors used a CD-38 based flow cytometry assay (page 10), but, following the literature, a commonly accepted marker profile for M1-macrophages is CD68+/CD80+, whereas M2-macrophages are characterized as CD68+/CD163+ (Khramtsova G et al 2009 and Gordon S 2003). Can authors justify their choice? Can they provide these further functional assays?

We provide proper reference to justify the CD-38 based flow cytometry assay to identify LPS-activated populations in the presence and absence of HGA-derived pigment (page 7, 2nd paragraph).

A final but relevant consideration: the authors propose here a highly controversial molecule to avoid the limits of natural and synthetic melanins, but non-specific melanin-based probes for optoacoustic imaging have been just recently synthesized (Liopo et al 2015, Longo et al 2017, Repenko et al 2015), with an excellent optoacoustic contrast and a detection limit of such linear melanin about two orders of magnitude higher than that of the gold nanoparticles. In virtue of this, the use of HGA derived polymer seems not to be a good choice. And again, none of these papers has been cited by the authors in order to compare different probes.

To give the reader an appreciation for our new OA macrophage label, we performed a comparison against commercially available SiAu-nanorods (Figure 1 e-g). Additionally, we added further citations to acknowledge the ongoing work being done to improve (melanin-based) nanoparticles e.g. in terms of solubility and cell/tissue tolerance (Review from Longo et al 2017). However, as our agents is – in contrast to NPs – a pigment that is not pre-synthesized (lot-to-lot variation) and phagocytosed by cells, but rather formed *in situ/in vivo*, we consider this to be a valuable extension to the growing palette of OA-labels, especially for macrophage visualization.

Concluding, there is a significant need for standardized procedures for characterizing this HGA-derived polymer, as well as for protocols for its use in animal models in order to improve multi-site comparison (mice, as previously reported, have an endogenous production of ascorbic acid contrasting ochronotic HGA-based pigment production and are not the best animal model to evaluate HGA-induced pigmentation).

We thank the reviewer for this important information – consequently this implies, that in case an HDP-labeled macrophage should expel some pigment, it would be rendered harmless due to the endogenous production of ascorbic acid in mice.

Furthermore, to date, no standard protocols have been established to assess long term stability, accumulation, biodistribution, and safety that are mandatory to promote translation between basic research and clinical applications.

We emphasize again that we do not aim at direct administration of HGA, nor at its clinical use. Macrophages are highly versatile cells and their behavior in health and various diseases is far from being fully understood. We anticipate its use for non-invasive macrophage observation in model organisms utilizing OA imaging.

Reviewer #2:

A. Summary of the key results as interpreted by the reviewer

The present study is the first to demonstrate the OA properties of HDP *in vitro* and *in vivo* and to perform pilot studies of HDP application for intracellular labeling of macrophages.

B. Originality and interest: if not novel, please give references

The manuscript describes the original research. The idea of using homogentisic acid derived pigment as a biocompatible nanoparticle to enhance contrast of optoacoustic imaging is novel.

C. Data & methodology: validity of approach, quality of data, quality of presentation

Methodology is HDP production is well described, with appropriate controls. However, the methodology of HDP characterization is weak. Even though authors state that the optoacoustic contrast is strong, there is no quantitative numbers of such contrast in terms of minimally detectable concentration, increase of optoacoustic contrast per unit volume of HDP, absolute amplitude of optoacoustic signal as a function of concentration.

We thank the reviewer for the comment and added additional quantification data *in vitro* (Figure 1) and *in vivo* (Figure 3 a-e). To give the reader an appreciation of the signal strength in comparison to established labels we further conducted several measurements additionally with commercially available silica coated gold nanorods (Figure 1 e and f). For the *in vitro* quantification we additionally measured different concentrations of HDP-labeled macrophages in blood-agar and agar phantoms (Suppl. Figure S11).

D. Appropriate use of statistics and treatment of uncertainties

This is an exploratory study, so statistically significant data are not presented. Authors presented only demonstration of feasibility using single experiments.

We therefore increased the animal number to n=4 independent experiments for the visualization of HDP-labeled macrophage recruitment towards IFN γ implants. Since each mouse has an IFN γ positive as well as a control implant without IFN γ , each animal/experiment contains an internal control (Figure 3 f-i). Furthermore, we repeated the premixed cell-matrigel implantations with different cell numbers for implantation as well as PBS controls. Concerning the *in vitro* measurements of HDP-laden macrophages (primary- and immortalized), we carried out multiple repetitions in MSOT, starting with separately cultivated cell cultures and with different concentrations of HGA; multiple phantoms using the same cell concentrations and multiple positions acquired across each phantom. We are very content with the level of reproducibility that we achieve in MSOT with our syringe-based phantom setup that was specifically designed for this and other cell-based studies. Standard deviations are presented in the corresponding Figures.

E. Conclusions: robustness, validity, reliability

This manuscript describes a pilot research studies. Therefore, conclusions are somewhat ostensible.

As stated above, we increase the n-number of the recruitment experiments (Figure 3f-i). Moreover, we significantly expanded the histological confirmation data (Figure 3f-i, Figure 4 and Suppl. Figures S10 and S14). Additionally, we repeated the complete RSOM experiments, improved many technical aspects to overcome challenges such as e.g. motion caused by the animal's breathing and are now showing *in vivo* imaging of HDP-laden macrophages at a depth of up to 1mm in RSOM. Furthermore, we demonstrate robust detection of single cells fully mixed into blood (25%)-agar phantoms at different concentrations (Figure 5 and Suppl. Figure S15).

F. **Suggested improvements:** experiments, data for possible revision

The results obtained *in vitro* should provide more details in terms of optoacoustic signal contrast.

Concentrations in the range of mM seem too high for physiologically relevant conditions. Optoacoustic signal amplitude is provided on a relative scale no comparison is provided with background absorption of blood. Authors need to provide absolute values and quantities relevant to optoacoustic images, such as optical absorption coefficient associated with specific concentration of HDP.

As described above we added a more in-depth quantification of the OA signal. Further, we performed a side-by-side quantification of macrophages labeled with HDP or Si-gold nanorods. We again emphasize that we **do not** directly inject HGA, but label the primary macrophages *in vitro* and inject them to the animals subsequently. Thus, the primary concentration used for labeling is not relevant for further application of the labeled macrophages to the organism. That the HDP-labeled macrophages are alive and maintain their full functionality we confirmed with extensive tests on their differentiation behavior (Suppl. Figures S5 and S7), Cytokine release (Suppl. Figure S7) as well as motility (Figure 1g), in addition to LDH release to fully confirm that HGA-derived pigment is non-toxic to macrophages (Suppl. Figure S7a). All assays prove that the macrophages are fully functional after HDP labeling.

Since we are not observing HGA as an individual chromophore we do not report absorption coefficients, but rather state the OA strength generated by labeling the cells with a specific concentration or incubation time of HGA. Moreover, HDP is formed within the cell and is as such hard to quantify differently than by its apparent OA signal level - which we argue is the most relevant consideration, in addition to above mentioned features, for the actual *in vivo* experiments.

Also authors obtained images of raster scan optoacoustic microscopy showing existence of optoacoustic contrast in cells, but *in vivo* optoacoustic imaging data is not convincing.

The RSOM measurements were completely redone and show now robust detection of single cells in blood phantoms as well as sub cutaneous visualization of cells injected from a catheter needle (Figure 5 and Suppl. Figure S15). While the experiments clearly show that the HDP-labeled macrophages generate sufficient signal to be visualized on a strong blood background as well as *in vivo* it is technically challenging to resolve single macrophages *in vivo* in RSOM due to the movement of the cells even after *s.c.*-injection into the connective tissue. Nonetheless, utilizing the catheter needle allowed us to precisely define the injection area and acquire comparative images before and after injection. Single cell *in vivo* imaging in the context of disease is envisioned, but beyond the scope of this manuscript.

Authors used MSOT system to image small portions of a mouse body, but did not show detectable accumulation of their HDP contrast agent.

We do not look for HDP accumulation in the mouse body, we rather aim at visualizing HDP-labeled macrophages. Thus, we laid out the side-by-side IFN γ recruitment experiments to always provide a direct comparison to the +IFN γ (bait implant) to a control implant without IFN γ . Since we can primarily detect labeled cells in the bait implants we regard this as a proof of the detectability as well as functionality of the HDP-labeled macrophages (Figure 3f-i). As stated above we linked our MSOT results with extensive histology (Figure 3f-i, Figure 4 and Suppl. Figures S10 and S14) confirming the presence of labeled (Schmorl's stain) macrophages (F4/80 staining). Moreover, we could attempt a level of co-registration between the MSOT signals and histology (Figure 4). We anticipate that our reviewers consider the fact that we are not imaging large tissue masses such as e.g tumors where cells are labeled for OA. We decided to challenge ourselves and therefore aimed at visualizing smaller cell densities *in vivo* that are the result of functional macrophage migration under a novel HDP-label.

... It seems like the contrast was not so great, so 3D images were not presented...

In fact, due to the small cell number the contrast cannot be compared with visualizing e.g. labeled solid tumors (e.g. synthesizing melanin, labeled with dyes or NPs)). To this end we increase the number of *in vivo* experiments in this study in combination with their histological verification.

G. References: appropriate credit to previous work?

The area of optoacoustic imaging research using various nanoparticles, including biodegradable

nanoparticles, has been of significant interest for almost two decades. However, authors refer mostly to their own works. Authors should search Scopus and other sources and provide relevant references that created basis of their own research and described significant accomplishments in the field nanoparticle enhanced optoacoustic imaging. As an example, this manuscript compares HDP with Eumelanin. However, they do not refer neither to the originally published photoacoustic images of genetically induced production of eumelanin nor melanin nanoparticles.

We added a paragraph focusing on the existing work on nanoparticles, in addition to adding SiAu-nanorods to our study (see above) and transgene expression of tyrosinase (page 2, 3rd paragraph).

We acknowledge the work that is done to improve solubility, cell compatibility and OA strength of melanin-based nanoparticles and cite a comprehensive review article.

However, we want to mention that macrophages are notoriously difficult to genetically modify, thus we see our approach of HDP labeling as an entry in the growing palette of OA-labeling agents and not in comparison with genetic systems. Lastly, our macrophage labeling method is not as expensive and time consuming as the generation of stable transgenic systems with which we do have extensive experience.

H. Clarity and context: lucidity of abstract/summary, appropriateness of abstract, introduction and conclusions

The abstract is formulated stronger than the results obtained. The conclusions need to be revised to state what was truly accomplished and remove forward looking statements.

With the addition of further experiments as requested by our reviewers, especially the *in vivo* and *in vitro* RSOM work, larger animal numbers and functionality assays for HDP-labeled primary macrophages, we think that the abstract is true to the data shown in this manuscript.

I. Minor edits

Replace “strong contrast in optoacoustic” with “strong optoacoustic contrast” in the Abstract.

We thank the reviewer for making us aware of this typo and changed it accordingly.

Reviewer #3:

The manuscript by Stiel and colleagues have reported a new approach for labeling macrophages using HDP as optoacoustic imaging contrast. This timely work is significant because labeling macrophages without altering their viability and functions have been a technical challenge. The capability of labeling macrophages using HDP for in vivo tracking of the immune responses will have broad impact for a wide range of fundamental studies, especially for cancer research. The combination of HDP, which is water soluble, with optoacoustic imaging is also novel because it will potentially enable whole-body small animal imaging of the macrophage migration, with high-resolution and high-sensitivity. The paper was well written and easy to follow with sufficient details. With that, I feel that the manuscript is exciting but premature for publication in Nature Comm. My major and minor concerns are listed below.

1. The first major concern is the lack of chemical and biological studies about HDP, which in some cases is a pathological byproduct in alkaptonuria. It raises concerns about the biosafety of using this pigment as the macrophage labeling without a thorough and rigorous investigation about the pigment itself. I agree that the manuscript has presented a large number of experiments showing that HDP-labeled macrophages have apparently normal morphology and functions, which, however, still falls short to make up the lack of knowledge about the HDP itself. This actually suggests to me that this manuscript is better to be combined with more fundamental studies about the chemical, optical (physical) and biological properties of the HDP.

We thank the reviewer for this comment. First, we want to make clear that we do not inject HGA in the organism which might lead to pathological effects as in the disease alkaptonuria, instead we labeled macrophages *in vitro* leading to their intracellular pigmentation, followed by *i.v.* injection in the animals. However, we added additional experiments to ascertain the functionality and safety of HDP-labeled macrophages, namely a comprehensive cytokine assay (Suppl. Figure S7b), LDH viability assay (Suppl. Figure S7a), and a motility assay (Figure 1g). All

those experiments suggest that labeled macrophages are not impaired in their functionality in our time window of observation. Moreover, we also accounted for a very unlikely, but potential harmful, effect of the labeled macrophages or released HDP to the whole organism by performing a SAA assay with serum of animals after HDP-laden cell injection (serum amyloid A protein, a marker of acute phase - and chronic inflammation).

We agree with the reviewer that the chemical and physical nature of the HDP-pigment is a fascinating research direction that however goes beyond our pilot study of establishing HDP as an OA label to visualize functionally intact macrophages *in vivo* in whole organisms. We further want to emphasize that we revealed some of the biological aspects by localizing the pigment to lysosomes (Figure 2) and linking the strong OA signal to a peak at ~650 nm that only appears for pigment polymerized with intact cells (Supplements).

2. The second concern is about the advantage of using of HDP as a marker for optoacoustic imaging *in vivo*. The authors in the introduction have pointed out that the cell viability is ‘greatly affected by random intracellular enrichment of eumelanin’. This statement is not consistent with the published results from the UCL group using tyrosine (Nature Photonics volume 9, pages 239–246 (2015)), in which the heavy expression of melanin does not seem to reduce the cell growth rates of multiple cell lines. While this might be a different case for macrophages, the authors need more evidence on this statement because genetically encoded contrast is clearly superior over the *ex vivo* labeling contrast. Additional experiments using eumelanin-expressing macrophages are needed for a fair comparison.

The K562 (human, bone marrow, chronic myelogenous leukemia) and 293T (human, kidney, HEK cell-like) cell lines used in Jathoul et al. are immortalized cell lines. In contrast to this, we aimed to work with macrophages due to their natural mobility *in vivo*. Consequently, it was not an option for us to immortalize a primary line followed by genetic alteration as it is notoriously challenging (personal experience). We opted for primary cells which allow us to analyze HDP-label formation during different stages of differentiation and activation. The latter can be quite challenging in the presence of various nanomaterials (cited in main text) but proved to work with HDP. Hence, we focused our full attention on an approach to quickly and effectively label primary macrophages to follow their functionality *in vivo*. Additionally, the transient label bears several advantages as described in the following comment.

3. Following the comment above, despite the good performance of HDP-labeling, its application may be greatly limited by the fact that *ex vivo* labeling of the macrophages is needed before these cells can be returned to the animals to participate in the normal immune process. I agree that the present work is already a step forward for imaging macrophages in action. The need for *ex vivo* labeling and cell culturing have moderately reduced its potential applications and thus impact.

Having access to a transient labeling system for macrophages allowing the production of OA-visible cells in less than a week is not necessarily a disadvantage, on the contrary. One must take into consideration that genetic mammalian cell systems are only of advantage to the user when utilizing clonal cell lines with stable, uniform and homogenous expression of transgenes. Again, creating such systems in general, but especially with immune cells such as macrophages, is very challenging and time consuming.

4. It is clear that high concentration HGA (more than 1 mM) is harmful to the cells. In this case, it is critically important to analyze the underlining mechanisms of HGA or HDP’s cell-toxicity. A viable cell is not necessarily functioning normally.

We absolutely agree with the reviewer and thus extended our controls for the well-being and functionality of macrophages with extensive tests on their differentiation behavior (Suppl. Figure S5 – S7), LDH - and Cytokine release (Suppl. Figure S7) as well as motility (Figure 1g). All assays, in addition to the *in vivo* recruitment experiments (n=4), verify that the macrophages are fully functional after HDP labeling.

5. Minor. In the plots showing the optoacoustic signals, ‘intensity’ should be ‘amplitude’ because it is acoustic signals.

We changed all occurrences to “OA signal”.

6. The single-cell visualization in Fig. 4c is not very convincing. First of all, the RSOM system should not be able to resolve single cells with a 50 MHz transducer if the light is not tightly focused. How was the single-cell visualization validated?

The RSOM system uses a custom Lithium Niobate (LiNbO₃) transducer that has a very particularly wide bandwidth that leads to unmet resolution to depth ratios without the need for focused illumination (Aguirre, 2019). Such unusually wide bandwidth (>>100%) allows the system to have a resolution well beyond the one obtained by the usual lead zirconate titanate (PZT) transducers with bandwidths around 70%. Further details can be found in (Aguirre, 2019) and (Omar, 2019). In brief: the transducer covers a frequency range that spans from 10 to 120 MHz (Aguirre, 2019; Omar, 2019; Schwarz, 2017) which in turn leads to an axial resolution of 8 μm and a lateral resolution of 30 μm. Given the fact that the average size of the macrophages is ~20 μm (measured by an automated cell counter), the axial resolution is more than enough to differentiate the macrophages. In the lateral direction the cells will appear as the point spread function of the system. Such high resolution is kept with very little variation down to 1.5 mm depth (Aguirre, 2019).

This finding is further confirmed by our clustering analysis of event sizes in the blood-phantom RSOM experiment. The results indicate a major homogenous population of signals with sizes of single cells (Figure 5 and Suppl. Figure S15).

See details on the RSOM data analysis procedure Supp. Figure S17 and S18.

Further, the unmixing of the signals into red (low frequency) and green (high frequency) channels is not clear to me. Why should the macrophage signals be high frequencies? What is the cut-off frequency between the red and green channels?

The red channel corresponds to the frequency band from 10 to 40 MHz and the green channel corresponds to the frequency band from 40 to 120 MHz. Generally, the smaller the object the richer the high frequency content of the optoacoustic signals. In the case of the macrophages, the main frequency content lies in the 40-120 MHz range. Since high frequencies are absorbed by tissue more efficiently than low frequencies, by reconstructing separately the low frequencies from the high frequencies we can deal better with noise (SNR is worse for high frequency) (Aguirre, 2019). Even though the high frequencies are absorbed more efficiently by tissue frequencies above 100 MHz can be clearly found in signals incoming from more than 1 mm depth in tissue (Schwarz, 2017; Aguirre, 2014; Schwarz, 2015).

7. Minor. The OA-microscopy is using a 20 MHz transducer but a 3 GHz DAQ card? Is this an overkill for the sampling frequency? What does 'exposure 2.5 ms' mean for OAmicroscopy?

The DAQ card installed in our system supports acquisition rates up to 3 GS/s, but in the shown measurements we set the rate to 200 MS/s - which is a moderate oversampling rate in order to get a better accuracy of the waveform sampling. We recorded traces of 500000 records in order to get a good averaging of the signal. Consequently, it took 2.5 ms to record those traces ($5E5 / 2E8 = 2.5E-3$). We have triggered a laser shutter in a synchronized manner with the acquisition of the acoustic signal, to ensure that the sample is only illuminated with the laser light for the duration of acoustic recording. Therefore, we call this 'exposure time' = duration of illumination and recording.

8. Figure 4A. The multi-spectral imaging by MSOT is interesting. The extracted HDP signals were extremely sharp in the images. Given that the spectral coloring effect is always an issue for optoacoustic imaging which may induce errors in spectral unmixing, what was exactly the method to extract the HDP signals, and how were the imaging results validated?

Due to their lack of spectral features melanin-like spectra are notoriously hard to unmix, Since, due to the imaging of immune cells we observe relatively low cell numbers this results in a comparably low label density. Hence, in the revision, we optimized our unmixing approach. In brief: i) In Matlab (R2018), we clustered the spectra of all pixels within the slices of interest within the boundaries of the mouse-body-ROI using a k-means clustering algorithm with a fixed number of bins (200, we tested different bin numbers but achieved no better results using more bins). ii) Based on the experimental setup - here bait-implant and control-implant - we split the data in the mouse-body-ROI along the median plane and selected only clusters that had a predominant

occurrence in one of the two areas (80 % > evenly distributed). I.e. only clusters that are “over represented” on one of either side are selected. iii) From this set of clusters those that exhibit a high variance in comparison to an ideal melanin-like spectra were discarded. In this step spectra that occur predominantly unilateral due to anatomic reasons but are not related with the label are discarded. iv) The pixels of the remaining clusters were binned and visualized after one cycle of pixel homogenization as sum-projection. For the corresponding spectra the mean has been calculated after normalization and shown along with the image data. See Suppl. Figure S16.

For validation all mice have been cryo-sliced and the ROIs (bait-, control- and premixed implants) analyzed using a F4/80 macrophage marker and Schmorl’s staining to detect melanin-like pigments. The histology controls are shown together with the imaging data in Figure 3 and 4, as well as in Suppl. Figures S10 and S14.

We argue that the analysis strategy is especially suitable for low and potentially disperse signals. It can be adapted to other questions by subdividing the imaging data and comparing the regions based on spectral clusters exploiting the full multi-spectral capacity of MSOT. Manual inspection of spectral clusters with prevalence in specific ROIs secures identification of even small spectral differences reminiscent of the presence of labels in this area. Moreover, due to primarily looking for spectral peculiarities the method is relatively robust against spectral coloring since it only looks for changes of the spectra introduced by the agent present in the imaging volume.

9. Minor. Please add colorbars to the figures when applicable.

We added colorbars accordingly.

REVIEWERS' COMMENTS:

Reviewer #1 (Remarks to the Author):

The manuscript is now improved and the execution of the experiments was performed more appropriately.

The authors now specified that the objectives of the manuscript was to propose an original system to label macrophages in vivo for OA while they agree that this system differs in several substantial points from the genetic disease alkaptonuria (AKU). This statement is of fundamental importance, since the HGA-derived pigment for optoacoustic imaging application could otherwise be associated to the etiological agent of AKU, therefore leaving room to incongruities about some reported data and physiological effect of the studied pigment.

I would like to ask the authors to be very clear about that and to add a comment in the Introduction section and in the Discussion, that they are aware of these considerations.

Moreover, authors must specify in the manuscript that this kind of label was used in mouse, an animal model not perfectly reproducing the human physiological situation (e.g. the autoproduction of vitamin C in mice can act as antioxidant in vivo). This may render the mouse able to counteract the dangerous effect of HGA in the organism and harmless for animals, but not for humans.

Finally, when the authors state:

"Our primary macrophages grown in the presence and absence of 0.5 mM HGA for 5 days had normal levels of IL-6 (Suppl. Figure S7). This cytokine must be elevated in order to enhance the production of SAA. To be safe, we directly measured SAA in above macrophages which had no detectable levels (page 8, 3rd paragraph)."

they should cite a paper where plasma SAA was found to be high in spite of normal IL-6 levels (Osteoarthritis Cartilage. 2018 Aug;26(8):1078-1086. doi: 10.1016/j.joca.2018.05.017. Epub 2018 May 29).

Reviewer #2 (Remarks to the Author):

Authors provided comprehensive revisions of their manuscript addressing all questions of the reviewers. In its revised form this manuscript can be accepted for publication.

Reviewer #3 (Remarks to the Author):

The revised manuscript has been significantly improved from the original submission. The authors have spent great efforts addressing the concerns and questions raised by the reviewers. I applaud for their careful studies in the revised manuscript. Many previously raised questions have been well addressed. In particular, the potential cell toxicity induced by HGA and HDP has been clarified by additional more cell assays, of course, when the HGA concentration is low. Nor the animals injected with the HDP-laden macrophages showed elevated inflammations through a serum test. Moreover, it is satisfying to see that the HDP-laden macrophages do not show obvious dysfunctions. All these safety measures have demonstrated the potential applications of the new labeling strategy for OA imaging of macrophages. It is therefore my opinion that the revised manuscript can be recommended for publication if the authors can further address my remaining suggestions below.

First, I would like to reiterate my opinion about the novelty of this manuscript. I have no doubt about the novelty of using HGA for macrophage labeling, which may provide a viable way to image the immune responses in vivo. However, the novelty on the OA imaging is minimal if any at all, given the more detailed information provided in the revision. I have the following reasons.

- a. The imaging systems (whole-body OA, and RSOAM) have been well demonstrated and characterized in previous publications by the authors.
 - b. The data processing methods rely on mostly manual or empirical approaches such as k-spectral analysis, which do not show superior performance than the authors' other works such as the eigenvector based approach.
 - c. With the strong and complex background signals in vivo, in particular, from hemoglobin in blood, the sparse and relatively weak signals from the HDP-laden macrophages may be easily shadowed, or result in low specificity. This has been clear in Supp. Fig 11, in which the blood-agar signals have significant impact on the HDP signal generation. The linear dependence of OA signals on the HDP concentration is no longer valid, which may possess a challenge for the spectral analysis down the road.
 - d. The results demonstrated by both whole-body OA and RSOAM are not strikingly impressive. The injected macrophages were concentrated just beneath the skin surface, which does not take advantage of the deep-penetrating OA imaging. An animal model with deep-seating targets that can attract the macrophage migration (e.g., a liver tumor treated by high-intensity ultrasound) would have been a much stronger proof-of-concept study.
- Therefore, it is my opinion that this manuscript is novel in labeling macrophages using HGA, but not on the OA imaging.

A second point is that I would like the authors to be more straightforward on what have been actually achieved in this manuscript and what can be the potential in the future. Sometimes it is not very clear by reading the manuscript. For example, reading the abstract may result in the impression that single macrophage imaging has been achieved in vivo, which, however, is not the case as demonstrated in Fig. 5. I actually don't know if the features pointed by the arrows are actually macrophages, since there are no validation or spectral information shown.

I also suggest the authors to tune down their claim about the inferior performance of using commercially available nanoparticles. There are numerous nanomaterials that can be used for labeling macrophages, and the authors only tried one of them (silica coated nanorods). Nanorods are notoriously unstable for OA imaging because of its phase (dimension) change under strong optical illumination. The conclusion based on this single comparison is at least a bit thin.

Reviewer #1:

The manuscript is now improved and the execution of the experiments was performed more appropriately. The authors now specified that the objectives of the manuscript was to propose an original system to label macrophages in vivo for OA while they agree that this system differs in several substantial points from the genetic disease alkaptonuria (AKU). This statement is of fundamental importance, since the HGA-derived pigment for optoacoustic imaging application could otherwise be associated to the etiological agent of AKU, therefore leaving room to incongruities about some reported data and physiological effect of the studied pigment.

I would like to ask the authors to be very clear about that and to add a comment in the Introduction section and in the Discussion, that they are aware of these considerations.

We believe the manuscript is very clear about HDP being an imaging label with no direct implications for AKU research – but merely inspired by said research. However, to make it even more clear we added a statement to the introduction and the discussion that we have no evidence that the macrophage label HDP is similar to the etiological agent of AKU (p3 3rd paragraph, p9 1st paragraph).

Moreover, authors must specify in the manuscript that this kind of label was used in mouse, an animal model not perfectly reproducing the human physiological situation (e.g. the autoproductioin of vitamin C in mice can act as antioxidant in vivo). This may render the mouse able to counteract the dangerous effect of HGA in the organism and harmless for animals, but not for humans.

As indicated in our previous response we did not observe any toxicity in the time window of observation. Moreover, the HDP-labeling will likely never be used with organisms that suffer from AKU. We believe that healthy organisms can tolerate low amounts of HGA should they arise. Beyond that we agree with the reviewer that in mouse the antioxidant effect of vitamin C could counteract negative long-term effects. Moreover, the labeling procedure is meant for life-sciences and not for immediate translational research, thus the applicability of this label in humans is out of the question.

Finally, when the authors state:

“Our primary macrophages grown in the presence and absence of 0.5 mM HGA for 5 days had normal levels of IL-6 (Suppl. Figure S7). This cytokine must be elevated in order to enhance the production of SAA. To be safe, we directly measured SAA in above macrophages which had no detectable levels (page 8, 3rd paragraph).” they should cite a paper where plasma SAA was found to be high in spite of normal IL-6 levels (Osteoarthritis Cartilage. 2018 Aug;26(8):1078-1086. doi: 10.1016/j.joca.2018.05.017. Epub 2018 May 29).

We thank the reviewer for that comment and added the citation (p6, 6th paragraph).

Reviewer #2:

Authors provided comprehensive revisions of their manuscript addressing all questions of the reviewers. In its revised form this manuscript can be accepted for publication.

We thank the reviewer for the stimulating remarks in the first revision that greatly helped improve this manuscript.

Reviewer #3:

The revised manuscript has been significantly improved from the original submission. The authors have spent great efforts addressing the concerns and questions raised by the reviewers. I applaud for their careful studies in the revised manuscript. Many previously raised questions have been well addressed. In particular, the potential cell toxicity induced by HGA and HDP has been clarified by additional more cell assays, of course, when the HGA concentration is low. Nor the animals injected with the HDP-laden macrophages showed elevated inflammations through a serum test. Moreover, it is satisfying to see that the HDP-laden macrophages do not show obvious dysfunctions. All these safety measures have demonstrated the potential applications of the new labeling strategy for OA imaging of macrophages. It is therefore my opinion that the revised manuscript can be recommended for publication if the authors can further address my remaining suggestions below.

We greatly thank the reviewer for these encouraging comments

First, I would like to reiterate my opinion about the novelty of this manuscript. I have no doubt about the novelty of using HGA for macrophage labeling, which may provide a viable way to image the immune responses in vivo. However, the novelty on the OA imaging is minimal if any at all, given the more detailed information provided in the revision. I have the following reasons.

a. The imaging systems (whole-body OA, and RSOAM) have been well demonstrated and characterized in previous publications by the authors.

The clear aim of the MS is to explore unique ways of providing contrast to macrophages – a cell type with a rather sparse set of existing labels. For the imaging we deliberately used tried-and-true and published OA setups.

b. The data processing methods rely on mostly manual or empirical approaches such as k-spectral analysis, which do not show superior performance than the authors' other works such as the eigenvector based approach.

The "Eigenvector based approach" (eMSOT) mentioned by the reviewer (Tzoumas, Nat Comm., 2016) is exclusively used for determining blood oxygenation levels and not meant for unmixing distributions of labels. It is important also to understand that the problem of locating a group of cells with a specific spectrum within tissue does not have anything to do with the problem of quantifying Oxy- and Deoxy-hemoglobin. While the first problem deals with segmentation, the latter is an ill-posed inverse problem. In detail, eMSOT uses pre-computed data based on simulating the convolution of Oxy- and Deoxy-Hemoglobin spectra in dependence of different light fluencies. Such data is not available for the spectral characteristics of label-compounds. This, however, does not mean that it is not theoretically possible to adapt the strategy in such ways, i.e. simulate fluency dependent convolutions of label spectra with Oxy- and Deoxy-Hemoglobin and other absorbers likely entangled with the labels' spectral signature. Though, it must be noted that eMSOT has to only consider a convolution of two spectra due to its very focused application (blood oxygenation), an extension of the concept towards involving more spectral signatures significantly increases the complexity of the approach. Nonetheless, we thank the reviewer for this inspiring thought and will discuss future adaptations of the eMSOT concept to label detection with the respective authors.

In general, the presented work is not meant to compare or develop unmixing algorithms but to introduce a prospective labeling concept for macrophages.

c. With the strong and complex background signals in vivo, in particular, from hemoglobin in blood, the sparse and relatively weak signals from the HDP-laden macrophages may be easily shadowed, or result in low specificity. This has been clear in Supp. Fig 11, in which the blood-agar signals have significant impact on the HDP signal generation. The linear dependence of OA signals on the HDP concentration is no longer valid, which may possess a challenge for the spectral analysis down the road.

Regarding Suppl. Fig. 11 we do not see an obstruction of the linear dependence of the signal in the blood-agar phantoms. In both cases, plain – as well as blood agar, the signal decreases linearly with cell concentration. However, we clearly agree with the reviewer, and have also stated so repeatedly in the MS, that the spectrum of melanin is notoriously hard to unmix. In general, label concentrations are always a tight balance between signal and functionality of the recipient cell, especially in regard to cells with

such delicate functionality as macrophages. Given those considerations we could clearly demonstrate visualization of different cell numbers, in phantoms and in vivo.

d. The results demonstrated by both whole-body OA and RSOAM are not strikingly impressive. The injected macrophages were concentrated just beneath the skin surface, which does not take advantage of the deep-penetrating OA imaging. An animal model with deep-seating targets that can attract the macrophage migration (e.g., a liver tumor treated by high-intensity ultrasound) would have been a much stronger proof-of-concept study. Therefore, it is my opinion that this manuscript is novel in labeling macrophages using HGA, but not on the OA imaging.

As pointed out above, we second the reviewer in the sense that the manuscript does not aim at developing novel OA imaging techniques but shows a novel and exciting pilot study for a prospective labeling approach. One, that upkeep high cell viability and shows no perturbation of the labeled macrophages function while simultaneously still ensuring polarization to pro-inflammatory cells. We demonstrate this by systemically injecting the labeled cells in mice and showing their functional recruitment to a defined area in the body. To our knowledge, such a challenging experimental setup has not yet been demonstrated for OA in vivo imaging, while utilizing pre-labeled macrophages.

Beyond that we do think that seeing cell populations non-invasively in the dermis at 1 mm depth below the skin using RSOM is a critical achievement beyond the reach of most optical approaches. It is important to note that the skin represents a double challenge for pure optical approaches. The problem is not only light scattering, also, the layered structure of the skin, each layer having a different index of refraction, imposes an extra restriction to optical approaches, since focusing is severely hampered. Moreover, numerous disease models are studied in the skin, e.g. orthotopically injected cancers etc. Not to mention major skin diseases related to inflammation, some of them like psoriasis, still have no cure, it is not well understood and it affects a great range of the population. In all these cases better visualization of components of the immune system like macrophages is a crucial aspect. We agree that liver is another highly relevant target to understand implications of the immune system but to our knowledge present OA technology is challenged by resolving smaller cell populations in the liver due to its exceptionally strong background from blood hemoglobin. In that sense we believe that future development in regard to OA instrumentation, image reconstruction and unmixing techniques will further help tackle such challenges and should go hand-in-hand with developments of novel labeling strategies – such as ours.

A second point is that I would like the authors to be more straightforward on what have been actually achieved in this manuscript and what can be the potential in the future. Sometimes it is not very clear by reading the manuscript. For example, reading the abstract may result in the impression that single macrophage imaging has been achieved in vivo, which, however, is not the case as demonstrated in Fig. 5. I actually don't know if the features pointed by the arrows are actually macrophages, since there are no validation or spectral information shown.

We would like to emphasize that the pre-injection image shown (Figure 5b and d) already serves as a strong control and the features highlighted appear only post-injection (Figure 5c and e) showing, very convincingly, a trail of high-absorbing macrophages. Note that no experimental change occurred in between the two consecutive time points except for the necessary injection of the labeled cells via the catheter tubing. Hence, changes due to potential hemorrhages are extremely unlikely as the needle was indeed already inserted at the pre-injection time point and only the labeled macrophages were released by the syringe piston before capturing the consecutive post-injection image. Nonetheless, we have added data at a control wavelength of 532 nm which shows the vessel structure due to a higher blood absorbance at this wavelength. It is apparent from the data that blood vessels present higher absorbance at 532 nm and are only faintly appearing at 630 nm, precisely expected from the spectral signatures of oxy- and deoxy-hemoglobin. Conversely, the signals we identify as HDP labeled macrophages show little difference in signal intensity at 532 and 630 nm, in agreement with melanin absorbance spectra. We added this data as Suppl. Figure 19a and b.

To further support our claim and for reviewing purposes, we provide here the histology of the measured mouse which clearly shows an enrichment of black mass at the tip of the needle already apparent to the naked eye (Suppl. Figure 19c).

Despite these observations and following the reviewer suggestion, we disentangled the statement of in vitro blood phantom single-cell resolution from the in vivo RSOM imaging in the abstract to avoid misconception. Although from the data it is clear that subcutaneous single-cell resolution is achievable and we showcase it as a proof of concept in the manuscript; a challenging next step is to track our label along with a dedicated RSOM device able to perform fast imaging to resolve mobile single cells in vivo. This represents an advanced but separate technical development beyond the scope of this work.

I also suggest the authors to tune down their claim about the inferior performance of using commercially available nanoparticles. There are numerous nanomaterials that can be used for labeling macrophages, and the authors only tried one of them (silica coated nanorods). Nanorods are notoriously unstable for OA imaging because of its phase (dimension) change under strong optical illumination. The conclusion based on this single comparison is at least a bit thin.

We clearly stated that this is just a single pick of commercially available nanoparticle which is however well used and suited. We are clear that the claims do not necessarily have to be true for nanoparticles in general. Although we think, especially since there is a debate regarding the bio-functionality of nanoparticles in the field it was important to share all our observations with the reader. A broader study on macrophage labeling techniques with a number of different nanoparticles would definitely be beneficial, should be undertaken in the future, but is not the focus of this work. Nonetheless, to further emphasize that our study only used one nanoparticle for comparison and does not speak for the wealth of existing nanoparticles we added the specific type again to the discussion (p9, 3rd paragraph).